EMBO
Molecular Medicine

# Affinity proteomics within rare diseases: a BIO-NMD study for blood biomarkers of muscular dystrophies

Burcu Ayoglu[1], Amina Chaouch[2], Hanns Lochmüller[2], Luisa Politano[3], Enrico Bertini[4], Pietro Spitali[5], Monika Hiller[5], Eric H Niks[6], Francesca Gualandi[7], Fredrik Pontén[8], Kate Bushby[2], Annemieke Aartsma-Rus[2,5], Elena Schwartz[9], Yannick Le Priol[10], Volker Straub[2], Mathias Uhlén[1], Sebahattin Cirak[11], Peter A C 't Hoen[5], Francesco Muntoni[12], Alessandra Ferlini[7], Jochen M Schwenk[1], Peter Nilsson[1] & Cristina Al-Khalili Szigyarto[13,*]

## Abstract

Despite the recent progress in the broad-scaled analysis of proteins in body fluids, there is still a lack in protein profiling approaches for biomarkers of rare diseases. Scarcity of samples is the main obstacle hindering attempts to apply discovery driven protein profiling in rare diseases. We addressed this challenge by combining samples collected within the BIO-NMD consortium from four geographically dispersed clinical sites to identify protein markers associated with muscular dystrophy using an antibody bead array platform with 384 antibodies. Based on concordance in statistical significance and confirmatory results obtained from analysis of both serum and plasma, we identified eleven proteins associated with muscular dystrophy, among which four proteins were elevated in blood from muscular dystrophy patients: carbonic anhydrase III (CA3) and myosin light chain 3 (MYL3), both specifically expressed in slow-twitch muscle fibers and mitochondrial malate dehydrogenase 2 (MDH2) and electron transfer flavoprotein A (ETFA). Using age-matched sub-cohorts, 9 protein profiles correlating with disease progression and severity were identified, which hold promise for the development of new clinical tools for management of dystrophinopathies.

**Keywords** antibody-based proteomics; disease severity biomarkers; Duchenne muscular dystrophy; plasma profiling; protein profiling

**Subject Categories** Biomarkers & Diagnostic Imaging; Musculoskeletal System; Systems Medicine

## Introduction

A plethora of proteomics tools, including mass spectrometry and affinity-based protein profiling approaches, is being increasingly applied for analysis of body fluids, which aim to reveal many new candidate biomarkers for diagnosis, prognosis, or surveillance of, for example, most common cancer types with high incidence rates. An equally urgent need for such protein markers exists also in rare diseases, which are defined as affecting one person in every several thousands or millions. Sample availability is, however, a limiting factor and the main impediment to progress of research in rare diseases, resulting in a remarkable lack of attempts to apply protein profiling approaches in the quest for protein markers in rare diseases.

One such example is Duchenne muscular dystrophy (DMD), which is a rare X-linked genetic disease with an incidence rate of 1:5,000 in Wales and 1:6,000 in Ohio, as estimated by screening of newborn male subjects (Kalman *et al*, 2011; Mendell *et al*, 2012;

1   Affinity Proteomics, SciLifeLab, School of Biotechnology, KTH-Royal Institute of Technology, Stockholm, Sweden
2   Institute of Genetic Medicine, Newcastle University, Newcastle upon Tyne, UK
3   Cardiomiology and Medical Genetics, Department of Experimental Medicine, Second University of Naples, Naples, Italy
4   Unit of Neuromuscular and Neurodegenerative Disorders, Department of Neurosciences, Bambino Gesú Children's Hospital, Rome, Italy
5   Department of Human Genetics, Leiden University Medical Center, Leiden, The Netherlands
6   Department of Neurology, Leiden University Medical Center, Leiden, The Netherlands
7   Unit of Medical Genetics, Department of Medical Sciences, University of Ferrara, Ferrara, Italy
8   SciLifeLab, Department of Immunology, Genetics and Pathology, Uppsala University, Uppsala, Sweden
9   Ariadne Diagnostics, Rockville, MD, USA
10  Elsevier, Amsterdam, The Netherlands
11  Research Center for Genetic Medicine, Childrens National Medical Center, Washington, DC, USA
12  The Dubowitz Neuromuscular Centre, UCL Institute of Child Health, London, UK
13  Department of Proteomics, School of Biotechnology, KTH-Royal Institute of Technology, Stockholm, Sweden
*Corresponding author. Tel: +46 8 5537 8832; Fax: +46 8 5537 8481; E-mail: caks@kth.se

Moat *et al*, 2013). DMD is caused by frame-disrupting mutations in the gene coding for dystrophin, resulting in loss of dystrophin. Affected boys typically present in the first few years of life with features suggestive of muscle weakness and often with global developmental delay. Progressive muscle weakness leads to loss of ambulation by the age of 10, and if untreated, to fatal cardiorespiratory insufficiency by the late teens. Becker muscular dystrophy (BMD) is the milder allelic form of the disease with an incidence rate of 1:20,000 (Bushby *et al*, 1991; Moat *et al*, 2013). BMD is characterized by mutations that leave the open-reading frame intact, resulting in the presence of internally deleted and often reduced levels of dystrophin (Mercuri & Muntoni, 2013). Males affected by BMD present later in life than those with DMD, mostly with a variable degree of exercise intolerance, and despite progression, BMD patients are usually able to remain ambulant until late in adult life.

Establishing a correct diagnosis in dystrophinopaties (DMD and BMD) requires a multidisciplinary approach involving pediatricians, geneticists and neurologists to define the severity of the clinical phenotype by means of genetic, enzymatic and immunohistochemical tests (Manzur & Muntoni, 2009; Bushby *et al*, 2010; Verma *et al*, 2010; Ferlini *et al*, 2013). Dystrophin is invariably absent on muscle biopsy from DMD patients, whereas BMD patient muscle biopsies show dystrophin albeit at reduced levels or in a mosaic pattern (Mercuri & Muntoni, 2013). Creatine kinase (CK) levels in blood, elevated for both DMD and BMD, are also indicative of muscle damage. However, it does not correlate well with disease severity, being influenced by multiple factors such as amount of muscle mass, age and level of physical activity (Malm *et al*, 2000; Baird *et al*, 2012). Currently, disease progression and response to potential treatment are monitored by clinical assessments via consolidated functional outcome measures and invasive testing using muscle biopsies. Muscle magnetic resonance imaging (MRI) is being under development and may hold promise as a noninvasive tool (Mazzone *et al*, 2013). However, MRI is expensive and not suited for young children unless sedated or anaesthetized, which is not desirable in muscular dystrophy. Given that multiple clinical trials are ongoing or planned in DMD (Rodino-Klapac *et al*, 2013), it is important to develop new outcome measures correlating with disease severity. Molecular biomarkers present in body fluids such as proteins or microRNA (Cacchiarelli *et al*, 2011a,b) would be ideal to monitor patient health status and, if validated, could also be used for disease and patient stratification for clinical trials.

To explore the possibility of identifying circulating candidate protein markers in rare diseases, we applied an affinity-proteomics approach to generate proteomic signatures in blood of muscular dystrophy patients and controls. A total of 345 blood samples collected by the EU-FP7 BIO-NMD consortium partners (www.bio-nmd.eu) from four clinical sites in Europe were profiled using a multiplexed antibody suspension bead array for 315 unique proteins targeted by 384 antibodies, all generated within the Human Protein Atlas (www.proteinatlas.org) (Uhlén *et al*, 2010; Fagerberg *et al*, 2013). Across the different center collections and blood preparation types, data analysis yielded eleven protein profiles that consistently differed between muscular dystrophy phenotypes and controls. These profiles belonged almost exclusively to proteins involved in muscle function and regeneration or are annotated as being specifically expressed in muscle tissue. These discovered protein profiles may serve as a starting point for development of clinical blood tests

to facilitate stratification and disease monitoring in dystrophinopathies, and the presented affinity-proteomics approach exemplifies a strategy toward the identification of blood-based protein biomarkers in rare diseases.

# Results

## Study and experimental design

We have here employed an affinity-proteomics approach using highly multiplexed antibody suspension bead arrays for proteomic profiling of serum and plasma samples. This setup is particularly favorable in the context of rare diseases affecting very young patients, as it requires only microliter amounts of sample material to generate protein profiles in serum and plasma. The concept of our approach is schematically presented in Fig 1A. The antibodies were coupled to color-coded magnetic beads, mixed to create an antibody array in suspension which was incubated with the nonfractionated, biotin-labeled samples to generate protein profiles in serum and plasma samples (Schwenk *et al*, 2008). The study was carried out with a focus on samples collected within the EU-FP7 BIO-NMD project (www.bio-nmd.eu). For this purpose, four clinical sites collected 345 samples from four different diagnostic categories of sample donors: DMD patients, BMD patients, healthy controls and asymptomatic female carriers (Fig 1A, Table 1). This geographically dispersed and, in this rare-disease context, large sample collection included various phenotypes, controls and blood preparation types. We performed both intra- and inter-cohort comparisons of samples from patients with different degrees of disease severity and focused especially on concordant protein profiles in the two different blood preparation types of serum and plasma and across the different cohorts (Fig 1B).

Regarding the design of the antibody array, a hypothesis-driven target selection approach was utilized, which was based on thorough mining of different data sources such as experimental evidence, pathway association, protein characteristics, and availability of validated antibodies within the Human Protein Atlas. For this purpose, a list of genes was compiled together with defined parameters including experimental evidence on protein or transcript level and/or the degree of association level of genes with muscular dystrophy. Existing experimental evidence on protein level by high-throughput LC-MS/MS techniques and immunoassays; on transcript level by RNA-seq; and on gene level by sequencing of single-nucleotide polymorphisms from analyses of samples from DMD patients with different phenotypes and/or response to steroid treatment was included. Values for each parameter were normalized on a scale of 0–1, each parameter was then multiplied by the given weight (40% for experimental evidence), and the weighted scores were summed. For each gene, additional parameters were included regarding their expression in normal muscle tissue estimated by immunohistochemistry; involvement in cellular pathways associated with muscular dystrophy, muscle contraction, sarcolemmal stability, and energy metabolism based on gene/protein annotation in curated databases; association to muscular dystrophy based on literature mining and accessibility in body fluids (Yuryev *et al*, 2006). As described above, values were attributed to each parameter and then weighted with 10% for all parameters except for the last

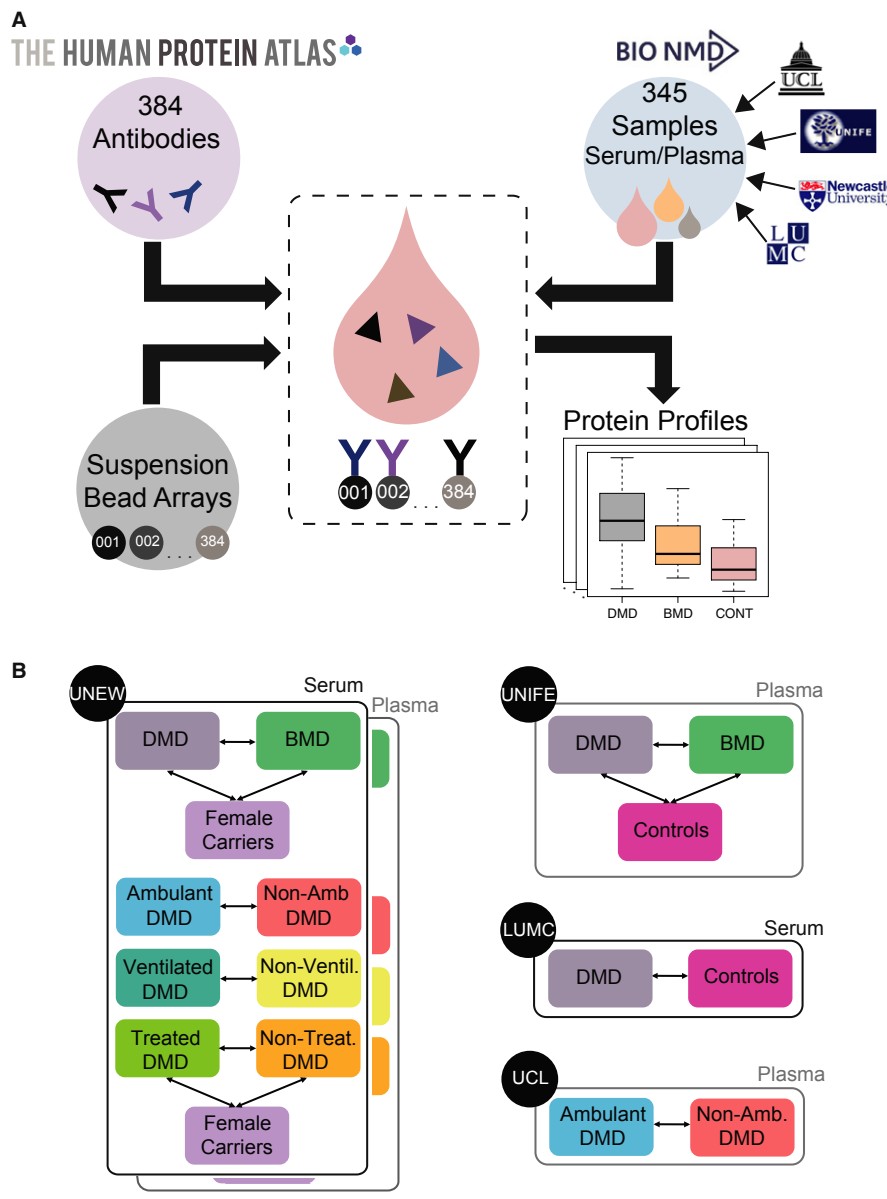

**Figure 1.  Overview of the affinity proteomics-based screening approach and study setup.**

A  An antibody suspension bead array platform was utilized to obtain profiles for 315 unique proteins in a total of 345 serum/plasma samples collected within the BIO-NMD project at different sites.

B  The captured proteins in serum/plasma using the antibodies generated within the Human Protein Atlas were analyzed and based on the obtained protein profiles, several inter- and intra-cohort comparisons in terms of diagnosis type and clinical parameters were performed.

one, which was weighted with 2.5%. Nine hundred and fifty-nine unique gene entries were finally ranked based on the summed weighted scores, and the top 315 genes with available validated antibodies from the Human Protein Atlas were selected. Only antibodies validated by Western blot and protein microarray were considered.

Information regarding technical aspects of the assay, such as the signal intensity distributions in serum and plasma (Supplementary Figs S1A and S2), the technical quality of the assay in terms of intra-assay % of coefficient of variations (CV) (Supplementary Fig S1B), the number of antibodies revealing signal intensities at noise level (Supplementary Table S1), and the number of correlating antibody

pairs targeting different parts of same protein in plasma and serum (Supplementary Fig S3), is available as supplementary information.

## Analysis of protein profiles within and across different cohorts

Analysis of the protein profiles from a global perspective by performing unsupervised hierarchical clustering of the entire dataset revealed that protein profiles were grouped mainly by blood preparation type (serum versus plasma) (Supplementary Figs S4 and S5A). Comparative analysis between disease and control groups was therefore carried out within each cohort and blood preparation type for identification of concordant differential protein profiles

**Table 1.  Overview of number of donor types and blood preparation types retrieved from different clinical sites.**

Samples from patients with BMD, DMD, healthy individuals (CONT) and female carriers of DMD and BMD (FC) collected at four different locations, that is, at Leiden University Medical Center (LUMC), University of Ferrara (UNIFE), University of Newcastle (UNEW) and University College London (UCL) were included in this study, resulting in a total of 345 samples (225 plasma and 120 serum preparations) collected at three different countries from 245 individuals.

| Diagnosis | Sample origin | Number of individuals | Sample type | Number of samples |
|---|---|---|---|---|
| DMD | UNEW | 60 | Plasma | 60 |
|  |  |  | Serum | 60 |
|  | LUMC | 12 | Serum | 12 |
|  | UCL | 40 | Plasma | 40 |
|  | UNIFE | 18 | Plasma | 18 |
|  | TOTAL | 130 |  | 190 |
| BMD | UNEW | 24 | Plasma | 24 |
|  |  |  | Serum | 24 |
|  | UNIFE | 9 | Plasma | 9 |
|  | TOTAL | 33 |  | 57 |
| FC | UNEW | 16 | Plasma | 16 |
|  |  |  | Serum | 16 |
|  | TOTAL | 16 |  | 32 |
| CONT | UNIFE | 58 | Plasma | 58 |
|  | LUMC | 8 | Serum | 8 |
|  | TOTAL | 66 |  | 66 |
| TOTAL |  | 245 |  | 345 |

(Fig 1B). The main contributors to the separation of patient and control groups in the different cohorts were proteins involved either in muscle-specific functions such as myosin light chain 3 (MYL3), calsequestrin-2 (CASQ2), microtubule-associated protein 4 (MAP4), or proteins highly expressed in muscle tissue such as carbonic anhydrase 3 (CA3) and malate dehydrogenase 2 (MDH2) (Fig 2). This supports the hypothesis of muscle proteins leaking into the bloodstream as a consequence of muscle wasting through disruption of the sarcolemma (Straub *et al*, 1997). Proteins involved in stress response such as matrix metalloproteinase 9 (MMP9), Parkinson disease protein 7 (PARK7) and proteins involved in metabolic processes such as rho-related BTB domain-containing protein 1 (RHOBTB1), creatine kinase (CK), electron transfer flavoprotein A and B (ETFA, ETFB) also contributed significantly to the discrimination between patients and controls.

Separation of DMD patients from controls was achieved within all cohorts, and the protein profiles of CA3 and MYL3 were the main common contributors (Fig 2). These two proteins together with CK, MDH2, and ETFA were main contributors for separating NMD patients and healthy controls in the UNIFE cohort (Fig 2A) and together with only MDH2 for the clustering of DMD serum samples and aged matched healthy donors in the LUMC cohort (Fig 2B). In the UNEW cohort, CA3 and MYL3 profiles contributed for clustering in both plasma and serum samples of DMD patients in comparison with female carriers. Comparison of the two muscular dystrophy phenotypes, DMD and BMD, in the UNEW cohort revealed again CA3

as an important contributor for the separation of these two patient groups (Fig 2C and D) and MDH2 and MYL3 for separation of the BMD patients and female carriers in both blood preparation types. Protein profiles that contributed to the clustering of patient groups in only one blood preparation type were also identified (e.g., CA3), which contributed for the clustering between BMD patients and controls in serum but not in plasma (Fig 2C and D).

## Protein profiles associated with muscular dystrophy

Differences in protein profiles revealed in both serum and plasma collected at different clinical sites are potentially more robust findings, for not being due to differences in sample preparation and handling. A nonparametric test was applied to identify protein profiles that were significantly different between any of the groups, that is, DMD, BMD, female carriers and healthy controls, and the resulting lists of proteins with $P$ values < 0.01 were compared, and the concordant findings in different cohorts were collected in Venn diagrams in terms of number of common proteins (Fig 3). Both for serum and plasma, levels for four proteins, CA3, ETFA, MYL3, and MDH2, were significantly different between DMD patients compared to controls, as shown in Fig 3A. These proteins allowed separation of DMD patients from both healthy controls and female carriers. Protein profiles for MDH2 and MYL3 could also separate between BMD patients and controls (Fig 3C), whereas CA3 allowed for separation of DMD and BMD patients from each other both in plasma and in serum (Fig 3E).

The classification performances of the identified concordant protein profiles were visualized by means of receiver operator characteristic (ROC) curves. The best performing protein panel consisting of CA3, ETFA, MYL3, and MDH2 had an area under the curve (AUC) ≥0.94 for classification between DMD patients and controls (Fig 3B). The dual panel of MYL3 and MDH2 gave AUC values of 0.77, 0.80, and 0.98 for the classification of BMD patients and controls in UNEW serum and plasma cohorts and in UNIFE cohort, respectively (Fig 3D). CA3 alone was also a good classifier for classification between the DMD and BMD patients especially for the UNIFE cohort with an AUC of 0.90 as compared to the UNEW cohort resulting in AUC values of 0.74 and 0.75 in plasma and serum, respectively (Fig 3F).

For these four proteins concordantly showing statistically significant differences, the distribution of MFI values across all individuals within the muscular dystrophy phenotype groups or controls is represented in Fig 4. CA3 was targeted in the assay by two different antibodies: CA3-Ab #1 raised toward the C-terminal part and CA3-Ab #2 raised toward the N-terminal part of the protein. The protein profiles generated by these two antibodies correlated well both in serum and in plasma (Spearman's ρ in serum = 0.83, in plasma = 0.80) (Supplementary Fig S3). Although the obtained signal intensity ranges differed, similar profiles were obtained for these two antibodies, with highest signal intensities in the DMD group, followed by the BMD group and with lowest signal intensities in the control groups. The other three proteins, MYL3, ETFA, and MDH2, revealing concordantly significant differences displayed the same trend as CA3, being more abundant in DMD patients than in BMD patients and with lowest levels in controls (Fig 4). Furthermore, the antibody pair targeting MDH2 was one of the 16 well-correlating antibody pairs (Supplementary Fig S3). We observed that the majority

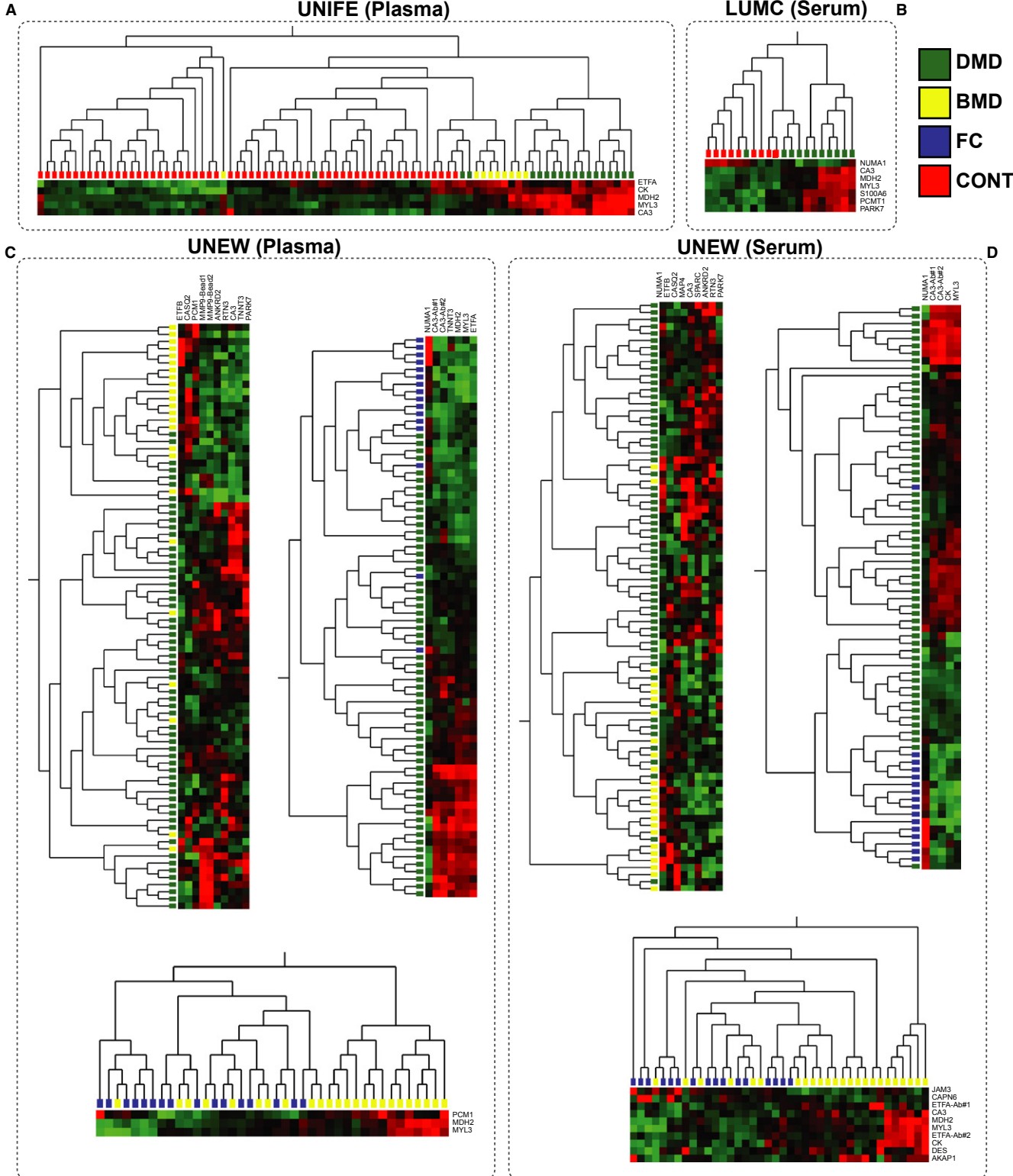

**Figure 2. Exploratory multi-protein profiles in plasma and/or serum of muscular dystrophy patients and control groups including healthy subjects or female carriers.**

A, C  Hierarchical clustering of protein profiles representing the main contributors for the grouping of plasma samples collected from DMD and BMD patients and controls at UNIFE (A) and UNEW (C).

B, D  Hierarchical clustering of protein profiles representing the main contributors for the grouping of serum samples collected from DMD and BMD patients and controls at LUMC (B) and UNEW (D).

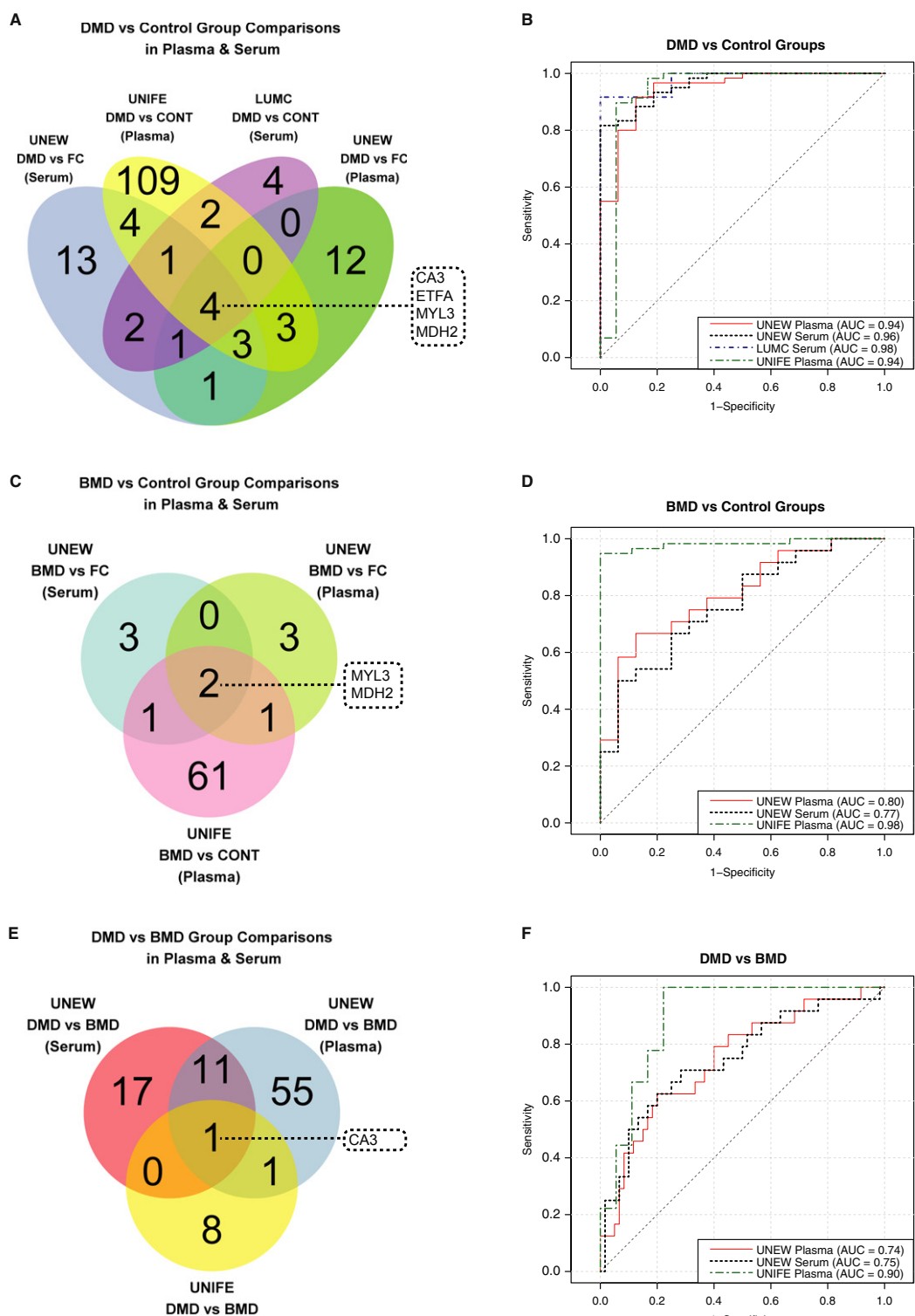

**Figure 3.    Identification and classification power of concordant protein profiles separating muscular dystrophy patients from control groups.**

A, C, E    Venn diagrams illustrate the number of proteins revealing significant differences (*P* value < 0.01) in different sample cohorts and blood preparation types for group comparisons between DMD patients and controls (A), BMD patients and controls and (C) DMD and BMD patients (E).

B, D, F    Classification power of these individual or combined protein profiles are represented on ROC curves. DMD patients and control groups were classified based on the panel composed of CA3, ETFA, MYL3 and MDH2 (B). A panel composed of MYL3 and MDH2 classified BMD patients and control groups (D) whereas CA3 alone classified DMD and BMD patients (F).

of the 'outliers' in these boxplots were BMD patients, in line with the generally higher degree of heterogeneity within BMD in terms of clinical phenotype as compared to DMD (Fig 4).

The comparison across DMD, BMD, and control groups revealed three more potentially interesting proteins, TNNT3, CK, and ETFB (also summarized in Table 2). Interestingly, the profile for ETFB showed an opposite trend (Fig 4) compared to other proteins including ETFA, which belongs to the same heterodimeric protein complex as ETFB. A negative correlation was revealed between the signal intensities for antibodies targeting these two proteins in DMD and BMD groups as compared to controls (Supplementary Table S2). The relationship between ETFA and ETFB was further analyzed by calculation of the ETFB/ETFA MFI value ratios for all the different sample groups and blood preparation types. Both in serum and in plasma, the ratios were highest in the control groups followed by the BMD patients and even lower in DMD (Supplementary Fig S6). Fitting a linear model revealed a statistically significant association between the change of ETFB/ETFA ratio and severity of the phenotype, in both serum and plasma (Fig 5).

Elevated levels of these seven proteins in blood from patients with muscular dystrophy can be explained by tissue leakage due to sarcolemmal disruption as a consequence of muscle contraction. To confirm that these proteins are present in healthy skeletal muscle and recognized by the antibodies used, immunohistochemical staining of tissues was performed (Supplementary Fig S7). The antibodies against CA3, MYL3, and TNNT3 selectively stained skeletal muscle, whereas those against MDH2, ETFA, and ETFB stained in addition various other tissue types (accessible through the Human Protein Atlas portal). In skeletal muscle, antibodies against CA3, MYL3, and TNNT3 stained the cytoplasm of myocytes, whereas those against MDH2, ETFA, and ETFB showed a granular cytoplasmic staining pattern, indicative of mitochondrial localization. MYL3 and TNNT3 showed strong staining of a subset of muscle fibers. The strong and muscle-specific staining of CA3, MYL3, and TNNT3 in healthy tissue indicates that detection of increased levels of these targets in blood samples of muscular dystrophy patients originates from the muscle, as these targets are not expressed in other tissues.

In order to further investigate the hypothesis that levels of muscle-specific proteins might be higher in serum/plasma of DMD and BMD patients, we dissected the protein profile trends across DMD, BMD, and CONT/FC groups for all of the muscle-specific targets included in our analysis. Out of 315 protein targets, 112 had been included in the study due to positive immunohistochemical staining of the antibodies in muscle tissue. These 112 'muscle-specific' proteins were targeted by 153 antibodies. Performing a SOTA cluster analysis across DMD, BMD, and CONT/FC groups within UNEW and UNIFE cohorts for each of these 153 protein profiles revealed protein profile trends being 'DMD increased' and/or 'BMD increased' as compared to the control groups (Supplementary Fig S8). In the UNIFE cohort, these were a total of 65 proteins (targeted by 73 antibodies), and in UNEW plasma and serum cohorts, there were a total of 62 proteins (targeted by 74 and 73

antibodies, respectively). The combined collection of these three sets of 'DMD/BMD increased' proteins consisted of 94 (out of 112) targets, and the intersection of these 3 sets consisted of 28 targets, including the above highlighted candidates MYL3, CA3, MDH2, ETFA, as well as dystrophin (DMD) or actinin 2 (ACTN2). Presumably, due to small sample sizes, not all of these protein profiles reached statistically significant difference levels concordant for both sample types in group comparisons. Nevertheless, almost 85% of them showed indeed in at least one cohort or sample type a protein profile trend, supporting increased leakage of muscle-specific proteins into circulation due to tissue damage in DMD and BMD patients.

## Association of identified protein profiles with disease development and clinical parameters

It is known that the health status of both DMD and BMD patients deteriorates with age, but at different rates. We addressed the question whether for any of the identified seven proteins, there is a correlation between protein levels and patient age and summarized this in Supplementary Table S3. Within the DMD group, for all targets except ETFB, there was a decrease with age. Particularly, the levels of MYL3, ETFA, and MDH2 revealed a strong decrease with age as compared to CA3, TNNT3, and CK (Supplementary Table S3). Within the BMD group, CA3 decrease did not correlate with age. Decreases in MDH2 and MYL3 correlated with age to a less degree in the BMD group in comparison with the DMD group. Most importantly, the correlations for these targets were much lower or close to zero in female carriers, and there was no correlation with age for any of the targets in the healthy controls.

Besides patient age, there are other hallmarks of disease progression and deterioration of muscular function that are rigorously monitored and used to assess health status of DMD patients. These include loss of ambulation, respiratory insufficiency, and cardiac dysfunction. Analyzing the protein profiles within the DMD cohorts by hierarchical clustering showed that the ambulant and non-ambulant DMD patients could be separated from each other (Supplementary Fig S9). Five of the seven previously mentioned proteins, CA3, MDH2, MYL3, ETFA, and TNNT3, contributed to the statistically significant discrimination of ambulant and non-ambulant DMD patients, both in serum and in plasma (Fig 6A). Additionally, a cytoplasmic protein, beta-enolase (ENO3), expressed in striated muscle tissue, revealed significantly different profiles between non-ambulant and ambulant DMD patients. The signal intensity levels for all of these six protein targets were decreased in non-ambulant patients in comparison with ambulant patients (Supplementary Fig S10A). Since the age distribution in the DMD cohorts was broad, we have defined and compared two smaller age-matched sub-groups of DMD patients from the UNEW cohort; one for ambulant patients and one for non-ambulant patients with mean age of 11.6 and 13.8, respectively. This comparison revealed the same trends for the six proteins, namely that the signal intensity

**Figure 4. Boxplots representing the seven protein profiles significantly differing between muscular dystrophy patients and control groups.**
Each boxplot represents the MFI values for CA3 (targeted by two different antibodies), MDH2, MYL3, ETFA, ETFB, TNNT3, and CK in plasma and/or serum of muscular dystrophy patients and control groups. Green and yellow boxes in different cohorts illustrate DMD and BMD patients, respectively, whereas the red boxes illustrate healthy controls and blue boxes the female carriers of DMD/BMD. For each sample group, the box-and-whisker plot represents MFI values within lower and upper quantile (box), the median (horizontal line within box), percentiles of 5 and 95% (whiskers) and outliers (dots).

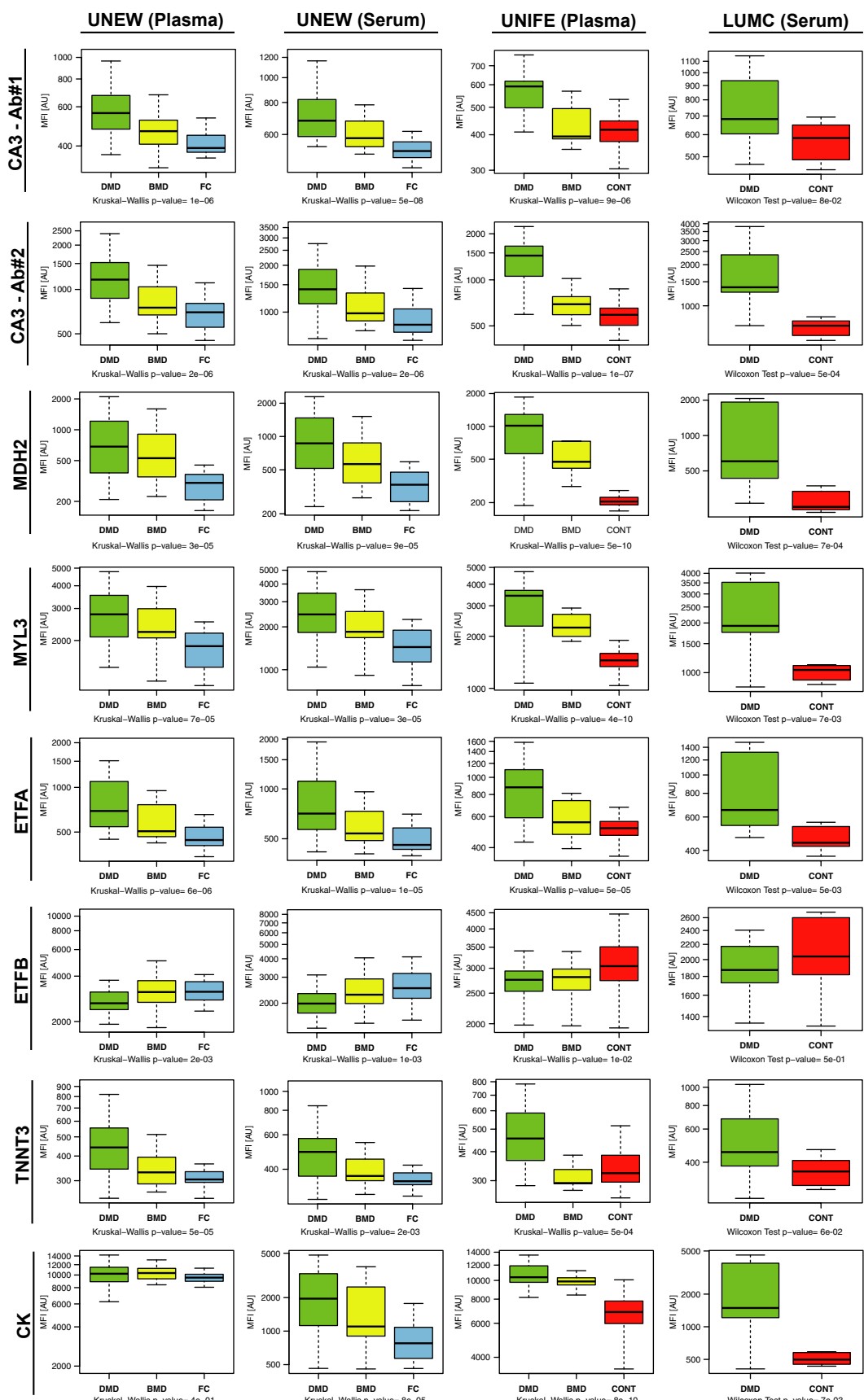

**Figure 4.**

levels decreased within the DMD patient group with loss of ambulation (Supplementary Fig S11). Next, the classification performance of these six proteins in serum and plasma samples of ambulant and non-ambulant UNEW and UCL cohorts was assessed. As indicated by the AUC values ≥ 0.91, sub-panels consisting of the seven antibodies targeting these six proteins allowed a good classification between ambulant and non-ambulant DMD patients (Fig 6B). Within the ambulant patient group from UNEW, the protein profiles showed little correlation with the NorthStar Ambulatory Assessment (NSAA) score (data not shown) (Mazzone *et al*, 2013). The limited number of patients (27) included in the analysis, and the complexity of the NSAA scoring system, which comprises assessment of 17 different activities related to gross motor function, might obscure subtle differences in protein profiles and highlights the need of detailed patient data recording in connection with sample retrieval.

We also compared the protein profiles for status of ventilation in order to identify protein profiles differing significantly between ventilated and non-ventilated DMD patients, all with an age above 14. Here, 16 ventilated patients (three ambulant and 13 non-ambulant) with an average age of 18 and 11 non-ventilated patients (all non-ambulant) with an average age of 21 from the UNEW cohort were included in the analysis. MDH2, ETFA, and TNNT3 revealed significantly higher MFI values in ventilated DMD patients as compared to non-ventilated DMD patients (Fig 6C and Supplementary Fig S10B). In addition to these three proteins, PPM1F (protein phosphatase 1F), COL6A1 (collagen alpha-1(VI) chain), and LCP1 (plastin-2) also contributed for the separation between ventilated and non-ventilated patients. Based on selected combination of smaller panels of these six targets, the ROC curve analysis resulted in AUC values ≥ 0.94 in plasma and serum (Fig 6D). Since for this analysis the number of patients in each group was relatively small, these protein profiles need to be further investigated in terms of their potential predictive value for respiratory dysfunction. We also did a similar comparative analysis between small groups of DMD patients with and without cardiac failure, but the statistical analysis revealed no concordant and statistically significant differences (data not shown).

One important factor, which potentially could influence the protein profiles in serum and plasma of DMD patients, is the treatment status of the patients. For instance, in the UNEW cohort, the majority of DMD patients were treated with steroids (deflazacort or prednisolone), whereas some had never been treated with any steroids. A multi-group comparison between these two sub-groups of patients and female carriers showed that the levels for CA3, MDH2, MYL3, ETFA, and TNNT3 were still elevated in both steroid-treated and non-treated patients in comparison with female carriers (Fig 7). Patient age distribution and variation in ambulation status make it yet difficult to investigate the effect of treatment outcome. Furthermore, lack of more detailed information about the patient response to treatment or treatment outcome measures highlights the need to collect longitudinal patient samples to investigate the further value of the identified proteins as clinical biomarkers.

## Discussion

In this study, we have investigated the levels of proteins in blood-derived samples of patients affected by the rare disease of muscular

dystrophy. The sample set was built from four cohorts collected at different clinical sites within the framework of the EU FP-7 project BIO-NMD. To determine the protein profiles in muscular dystrophy patients and controls, a high-throughput and multiplexed antibody suspension bead array setup was used in combination with antibodies generated by the Human Protein Atlas. In the context of muscular dystrophy, our study comprised a large set of samples, and among the 315 proteins studied, eleven were identified as potential candidates for discriminating between controls and muscular dystrophy patients and/or between the different phenotypes of muscular dystrophies, as well as between patients demonstrating different degrees of disease progression.

Protein biomarker discovery studies in blood-derived samples generally rely on findings revealed in one blood preparation type by screening a sample cohort from a single collection site. Yet, as demonstrated previously (Schwenk *et al*, 2010; Qundos *et al*, 2013; O'Neal *et al*, 2014) and in the presented study, different blood preparation types cause proteins to be detected differentially. Furthermore, despite the use of standardized sample collection, handling and storage protocols, it is difficult to retain identical conditions at and during transport from different clinical sites, where even subtle fluctuations might cause variations in the downstream analysis. To exclude the possibility that the findings are associated with a cohort of a specific origin or limited only to one blood preparation type, it is very valuable to base the findings on independent sample cohorts collected at different clinical sites, which ensures robustness of the findings at a very early stage. In line with this, the protein profiles we have denoted here as candidates were mainly selected and highlighted based on concordance of statistically significant differences revealed for these targets in more than one sample cohort and in both blood preparation types.

The utilized assay platform offers the possibility of generating hundreds of protein profiles in hundreds of patient samples in a single analysis, allowing for an effective exploration of potential candidates. The semi-automated workflow we developed for antibody coupling allows a very time-efficient generation of bead arrays consisting of up to 384 antibodies. While consuming only few microliters of crude plasma sample for a multiplexed profiling of hundreds of proteins, the lower limit of detection of this assay setup is in the higher pg/ml to lower ng/ml range (Schwenk *et al*, 2010). This allows not only for detection of proteins expected to be in plasma, such as CA3 with an average concentration of approximately 10 ng/ml (Mokuno *et al*, 1985; Ohta *et al*, 1991) or CK with a reference limit around 3–5 ng/ml (Apple *et al*, 2003), but presumably also for analysis of leakage products (Anderson & Anderson, 2002) from the muscle tissue.

The missing dystrophin has been shown to affect the composition of the muscle proteome, in particular the abundance of proteins involved in energy metabolism, muscle fiber contraction and stress response (Gardan-Salmon *et al*, 2011; Guevel *et al*, 2011; Carberry *et al*, 2012; Holland *et al*, 2013). Furthermore, due to its association with the transmembrane glucoprotein complex and its function, dystrophin when absent impairs the link between the intracellular contraction apparatus and the plasma membrane (Le Rumeur *et al*, 2010). Disruption of this link causes leakage of proteins from muscle fibers as a consequence of sarcolemmal damage during muscle contraction (Zweig *et al*, 1980; Hutter *et al*, 1991; Menke & Jockusch, 1995; Straub *et al*, 1997; Rando, 2012) and/or dysregulation

**Table 2. Summary of eleven identified blood marker candidates within muscular dystrophies and their level of statistical significance within various group comparisons.**

For each protein target, *P* values < 0.01 revealed in group comparisons are highlighted in light blue and *P* values < 0.001 are highlighted in dark blue.

| Antibody name | DMD versus control groups — Plasma UNEW | DMD versus control groups — Plasma UNIFE | DMD versus control groups — Serum UNEW | DMD versus control groups — Serum LUMC | BMD versus control groups — Plasma UNEW | BMD versus control groups — Plasma UNIFE | BMD versus control groups — Serum UNEW | DMD versus BMD — Plasma UNEW | DMD versus BMD — Plasma UNIFE | DMD versus BMD — Serum UNEW | AMB versus non-AMB DMDs — Plasma UNEW | AMB versus non-AMB DMDs — Plasma UCL | AMB versus non-AMB DMDs — Serum UNEW | VENT versus non-VENT DMDs — Plasma UNEW | VENT versus non-VENT DMDs — Serum UNEW |
|---|---|---|---|---|---|---|---|---|---|---|---|---|---|---|---|
| MDH2 | ● | ● | ● | ● | ● | ● | ○ |  |  |  | ● | ● | ● | ● | ● |
| MYL3 | ● | ● | ○ |  | ● | ● | ○ |  |  |  | ● | ● | ● |  | ● |
| ETFA | ● | ● | ○ |  | ● | ● |  | ○ |  |  | ● | ○ | ● | ● | ● |
| CA3-Ab#1 | ● | ● | ● |  | ● | ● | ○ | ○ |  |  | ● | ● |  |  |  |
| CA3-Ab#2 | ● | ● | ● |  | ● | ● |  | ● | ● |  | ● | ● |  |  | ● |
| TNNT3 | ● | ● | ○ |  |  |  |  | ○ |  | ○ | ● | ● |  |  | ● |
| ETFB | ○ |  |  |  |  |  |  |  |  |  | ● |  | ○ |  | ● |
| CK |  | ● | ○ |  |  | ● |  |  |  |  | ● | ● | ● | ● | ● |
| LCP1 |  |  |  |  |  |  |  |  |  |  | ● | ● |  |  |  |
| ENO3 |  |  |  |  |  |  |  | ○ |  |  | ● | ● |  |  |  |
| PPM1F |  |  |  |  |  |  |  |  |  |  |  |  |  | ○ |  |
| COL6A1 |  |  |  |  |  |  |  |  |  |  |  |  |  | ○ | ● |

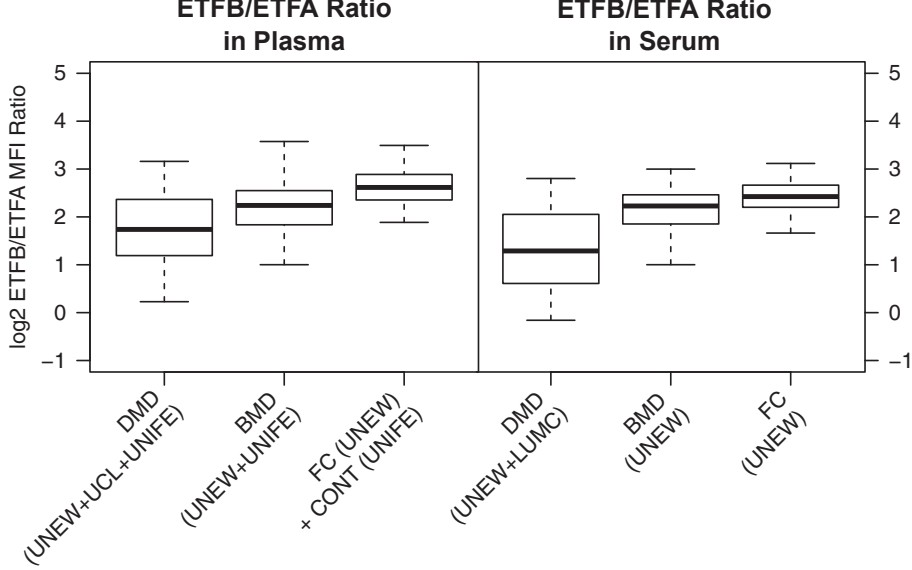

**Figure 5. Varying ETFB/ETFA ratios in plasma/serum of muscular dystrophy patients and control groups.**

A linear model was fit to the ETFB/ETFA level ratio in plasma/serum of muscular dystrophy patients and control groups, revealing a statistically significant association between the change of ETFB/ETFA ratio and the different diagnosis categories in serum (*P* value for linear model = 3e-07) and plasma (*P* value for linear model = 1e-14).

of the vesicular transport (Duguez *et al*, 2013). We therefore hypothesized that muscle proteins released into the bloodstream could function as indicators of patient phenotype and/or disease severity in muscular dystrophies. As hypothesized, we observed that 85% of the antibodies toward muscle-specific proteins revealed in at least one cohort or sample type a protein profile

trend supporting increased leakage of muscle-specific proteins into circulation due to tissue damage in DMD and BMD patients as compared to controls. Presumably due to small sample sizes, not all of these protein profiles revealed statistically significant differences concordant for both sample types in group comparisons. Yet, statistically significant and concordantly elevated levels were

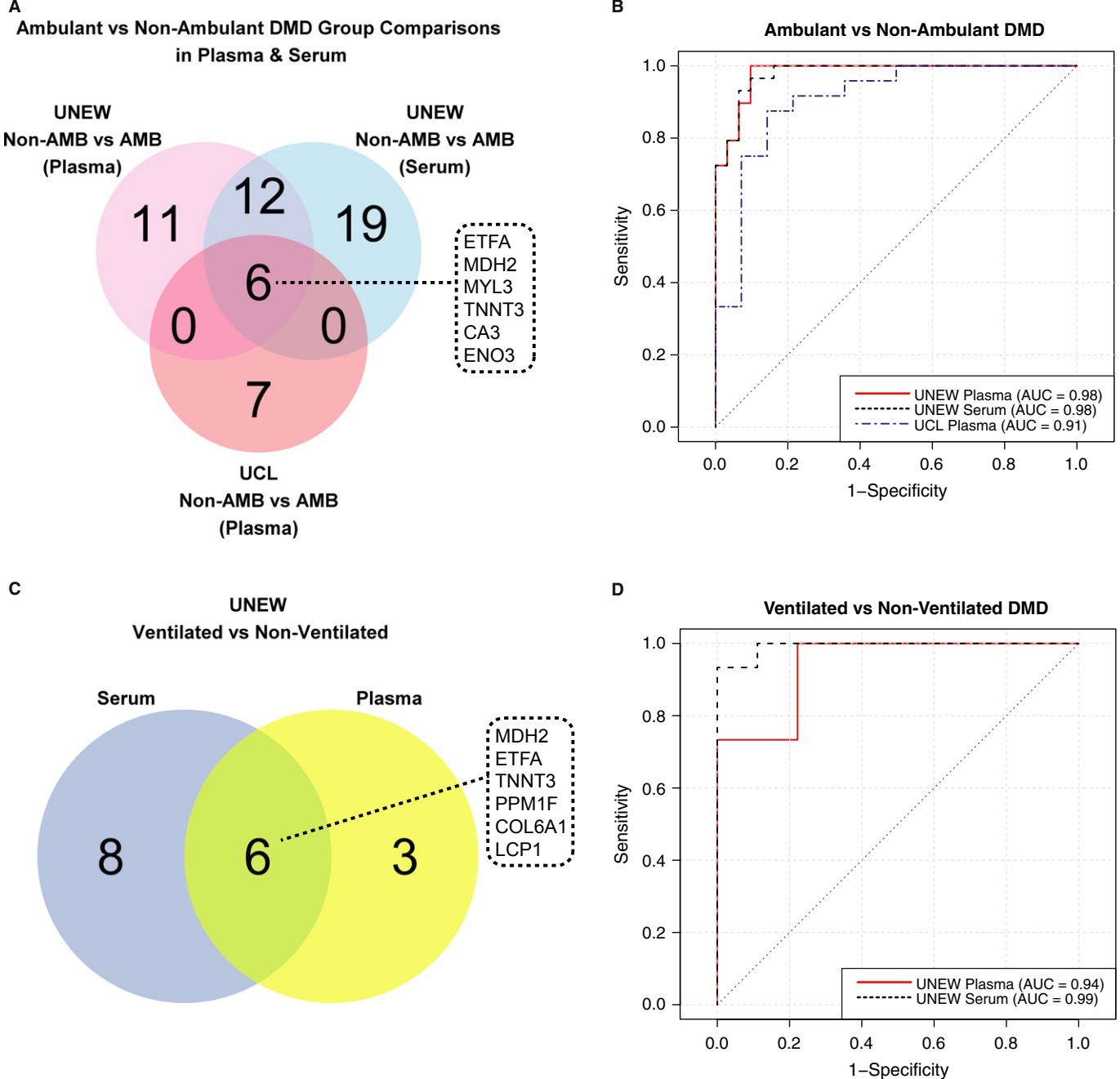

**Figure 6.  Association of protein profiles with ambulation status and respiratory insufficiency.**

A, C   Venn diagrams illustrate the number of proteins revealing significant differences (*P* value < 0.01) in group comparisons between ambulant (AMB) and non-ambulant (Non-AMB) DMD patients (A) and in ventilated and non-ventilated DMD patients (C).

B, D   ROC curves represent the classification power of a panel composed of CA3, ENO3, ETFA, MDH2, MYL3, and TNNT3 profiles between ambulant and non-ambulant DMD patients in UCL plasma cohort and UNEW plasma and serum cohorts (B). Classification of UNEW ventilated and non-ventilated patients was based on a panel including MDH2, PPM1F, ETFA, and TNNT3 for plasma and a panel including MDH2, PPM1F, COL6A1, and LCP1 for serum (D).

revealed for CA3, MYL3, and MDH2 in serum and plasma of DMD patients compared to healthy controls or female carriers. In comparison with these proteins, levels of CK, which has been used for confirmation of diagnosis of muscle wasting diseases for several decades (Mendell *et al*, 2012), were also found to be increased in patients, but with less degree of concordance and

statistical significance, yet still supporting the technical validity of our approach.

The proposed mechanisms by which these muscle proteins are released into the bloodstream are based on impaired secretion and/ or tissue leakage (Lippi & Banfi, 2009; Brancaccio *et al*, 2010; Duguez *et al*, 2013). *Mdx* myotubes have been reported to release

proteins such as MYL1 and MYL3 by lysosomal-associated membrane protein, LAMP1 vesicle mediated export due to an impaired secretion mechanism (Duguez *et al*, 2013). In contrast, CA3 has not been proven to be secreted through vesicle and could still enter the blood stream through tissue leakage due to membrane disruption. To elucidate the mechanisms by which the identified markers enter the blood stream requires additional experiments to establish in which way the transport is achieved. This could add great value to the understanding of the pathophysiological changes undergoing in muscles and the evaluation of these markers for clinical use.

Among the highlighted candidates, CA3 had been reported two decades ago to be elevated in serum of DMD patients but has not been re-evaluated since then (Carter *et al*, 1983). Our data further showed that CA3, which was targeted by two different antibodies in our assay, discriminated not only between DMD patients and healthy controls but also between the DMD and BMD clinical phenotypes. This protein has recently been reported to be increased in muscle of dystrophic chicken as compared to normal muscle (Nishita *et al*, 2012). Interestingly, carbonic anhydrase inhibitors have shown to have positive effects in animal models of dystrophin-opathies, suggesting they might be potentially explored for human therapy (Giacomotto *et al*, 2009). Expression of CA3 in different tissues, muscles and types of muscle fibers might explain the origin of the target detected in the bloodstream. CA3 is expressed in few tissues and therefore considered to be a more specific and sensitive marker for muscular dystrophies in comparison with CK, which is more ubiquitously expressed (Shima, 1984; Väänänen *et al*, 1988; Harju *et al*, 2012). CK abundance into the blood does not correlate with deterioration of specific muscle fibers upon tissue damage (Shima, 1984; Osterman *et al*, 1985; Väänänen *et al*, 1988), whereas CA3 does. CA3 is preferentially expressed at high levels in type I muscle fibers and is considered to be a marker for type 1 muscle fibers deterioration (Shima *et al*, 1983; Brancaccio *et al*, 2010). Consequently, elevated CA3 serum levels might reflect deterioration of skeletal muscles enriched in type I muscle fibers, such as the soleus muscle with a high CA3 content, rather than muscles enriched in type IIa and IIb fibers with low CA3 content (Frémont *et al*, 1988). The soleus muscle in patients with DMD is hypothesized to exert more power during ambulation, which causes hypertrophy or pseudohypertrophy and decrease in muscle mass (Cros *et al*, 1989). Studies in healthy individuals showed that CA3 is accumulated into the blood as a consequence of skeletal muscle injury (Väänänen *et al*, 1990; Brancaccio *et al*, 2010) and vigorous physical exercise (Takala *et al*, 1989) in a similar way as CK. Since continuous stimulation of muscles leads to increased expression of CA3 mRNA, it is difficult to conclude whether the elevated serum levels in NMD patients are due to muscle fiber replacement with connective tissue, increased expression of CA3 or both (Brownson *et al*, 1988). However, increased tissue expression due to other pathological conditions than muscular dystrophy, such as aging (Staunton *et al*, 2012), obesity and treatment with insulin or leptin (Alver *et al*, 2004), might affect serum levels of CA3. Thus, each and one of these effectors must be considered during follow-up studies.

There are several lines of evidence suggesting that the energy metabolism of dystrophic muscle is disturbed in DMD (Ikehira *et al*, 1995), which might explain the elevated ETFA levels in serum and

plasma of DMD patients. In contrast, ETFB, interacting with ETFA to build a complex involved in electron transfer from mitochondrial flavin-containing dehydrogenases to the respiratory chain, showed an opposite trend. ETFB was the only protein among the identified markers that revealed decreased levels in both plasma and serum of DMD patients as compared to BMD patients and controls. Our findings regarding the negative correlation between ETFA and ETFB and the gradual change of ETFB/ETFA ratio in DMD and BMD patients and controls could be the consequence of a relation between the accumulation of ETFA and dissipation of ETFB in patients with DMD and BMD as compared to controls. Yet, this opposite trend in abundance of ETFB and ETFA has not been reported previously, and the underlying reason for this finding cannot be explained at this point and remains to be investigated further.

Abundance of three serum proteins MMP9, metalloproteinase inhibitor 1 (TIMP1), and fibronectin (FN1) has very recently been shown to correlate with disease progression in patients affected by muscle dystrophy (Brancaccio *et al*, 2010; Nadarajah *et al*, 2011; Martin *et al*, 2014). In addition, potential protein markers for treatment outcome were identified in *mdx* mice plasma, for example, the coagulation Factor XIIIa (FXIIIa), leukemia inhibitory factor (LIFr), glutathione peroxidase 3 (GPX3), apolipoprotein E α (ApoE α), and β actins (ApoE β) (Alagaratnam *et al*, 2008; Colussi *et al*, 2009). Among these previously reported protein biomarkers, MMP9 was included in our target list (Nadarajah *et al*, 2011); however, the elevated levels of MMP9 in DMD patients compared to BMD patients were revealed only in plasma samples from a single clinical site and could not be confirmed in serum or in the other plasma cohorts.

Duchenne and Becker muscular dystrophies are characterized by progressively impaired muscular function with increasing age and muscular wasting. The majority of the protein profiles identified in this study showed a decrease with age in DMD patients, whereas only MDH2, MYL3 and ETFA showed a decrease with age in the BMD patients, although the age range in the latter group was wider. The rapid disease development and progressive muscle weakness experienced by DMD patients might be reflected in the strong decrease, considering the fact that there was no difference due to age in the control group. Nevertheless, this still highlights the inherent difficulty of distinguishing the effect of age and the effect of disease progression on protein levels in muscular dystrophies, underlining the need of ideally recruiting age-matched control subjects.

Establishment of the clinical phenotype, monitoring of disease progression, and disease management in muscular dystrophies involve assessment of motor functions, such as ability to walk and climb and assessment of respiratory capacity and cardiac function. We performed comparative analyses based on these clinical parameters to identify protein profiles potentially associated with such parameters. Within the DMD patient group, besides CA3, ETFA, MDH2, and MYL3, two other proteins, TNNT3 and ENO3, revealed significantly different profiles between the ambulant and non-ambulant DMD patients. Both of these proteins are muscle-specific proteins, ENO3 being involved in muscle development and regeneration (Ohara *et al*, 2006), whereas TNNT3 is involved in striated muscle contraction. Expression of TNNT3 has been previously reported to be decreased in muscle of dystrophic dog in comparison with healthy dogs (Gomes *et al*, 2004; Guevel *et al*, 2011). Considering the comparative analysis regarding ventilation in the UNEW DMD cohort, the proteins LCP1, COL6A1, and PPM1F together

with TNNT3, ETFA, and MDH2 exhibited decreased levels in both serum and plasma of non-ventilated DMD patients in comparison with those using a respiratory aid. COL6A1 has previously been linked to muscle regeneration and associated with other myopathies (Urciuolo *et al*, 2013), whereas LCP1 is a bundle protein linking actin filaments together and associated with autoimmune disease (Delanote *et al*, 2005). It should be noted that, while increased levels of skeletal muscle proteins in blood-derived samples indicate an increased muscle fiber breakdown and tissue leakage, increased levels of proteins, such as COL6A1 and LCP1, might be related to increased protein expression possibly indicating augmented connective tissue remodeling and inflammation. Interestingly, patients with respiratory insufficiency have lower levels of TNNT3, MDH2, and ETFA than the ones without, suggesting that patient sub-groups within the DMD cohort could be identified.

The presented discoveries from our screening efforts require further translation into assay systems that can be used in a clinical environment. This not only includes to develop a clinically robust test, preferably in a sandwich immunoassay format, but also to challenge the clinical sensitivity of this test with independent sets of samples. While the latter certainly is a challenge for rare diseases, we have yet recently shown a path of successfully translating 'discovery' assays into clinically more applicable tests (Qundos *et al*, 2014). Such a path would include (i) collecting commercially available mono- and polyclonal antibodies and generating monoclonal antibodies for targets with no commercial antibody availability, (ii) epitope-mapping of these antibody collections on high-density peptide arrays (Buus *et al*, 2012), and (iii) testing multi-antibody sandwich assays to identify matching pairs of these antibodies with distinct epitopes revealing a good assay sensitivity. From a clinical point of view, the most urgent need is to monitor disease progression in dystrophinopathies. Besides, there are also intermediate cases of DMD/BMD patients where genetic testing does not provide a clear diagnose. Therefore, although all the eleven candidates we have highlighted merit further investigation, initially, a marker panel consisting of CA3, MDH2, MYL3, TNNT3, and ETFA could be selected for development and further challenging with new sample material of such a 5-plex sandwich assay system as profiles for this set of five proteins would allow for an assessment of both the DMD/BMD and the ambulation status.

Although the candidate protein profiles we have highlighted here are not constrained by a certain blood preparation type, subsequent analyses of larger sample cohorts might indeed unveil whether any of these two blood preparations is preferable. For a future implementation of a marker panel, particular sample collection protocols and guidelines need therefore to be assessed and defined in order to achieve best and most robust assay performance. Such guidelines could then of course be applied across different cohorts and international study sites. Besides, levels of muscle proteins in the blood might be influenced by many parameters, such as overall muscle mass, amount of ongoing necrosis, and levels of physical activity as reflected by decreasing levels of muscle proteins in blood from older DMD patients. For future efforts in collecting and screening muscular dystrophy-related sample cohorts, it is crucial to clinically stratify patient cohorts according to age in combination with clinical phenotypes such as status of ambulation and/or respiratory function to fully elucidate these relationships. In addition, longitudinal studies in carefully characterized patient cohorts would support the evaluation of the utility of the identified protein profiles. Also, a matched analysis of muscle fibers and tissue with blood-derived samples would contribute for a better understanding of the origins and mechanisms leading to the proteins being present in blood. Matched tissue and blood samples are though not routinely collected and will require initiation of new sampling efforts with consent of patients and their guardians. Taken together, the presented study provides an important starting point for even more dedicated efforts within the muscular dystrophy community that will aim at elucidating disease pathogenesis further in multi-disciplinary collaborations including several centers with an ultimate aim of developing translated assays and assessing their performance at multiple clinical sites.

In conclusion, using an antibody-based proteomic profiling approach to screen geographically dispersed and independent cohorts of muscular dystrophy, we were able to identify proteins in blood that are involved in muscle function and energy metabolism. This demonstrated that other proteins than CK can be found accumulated in blood presumably as a consequence of muscle fiber injury or tissue leakage in muscular dystrophies. Our quest for potential, easily accessible blood marker candidates revealed presence of proteins that were indicators of disease phenotype and severity, making them key candidate proteins for novel clinical tests for diagnosis and management of muscular dystrophies. Furthermore, our approach demonstrated the possibility of gaining new insights into proteins altered in the circulation of patients with rare diseases even when only limited number and volumes of samples are available. Therefore, the affinity-proteomics approach we presented here offers a great promise for many other rare disorders with an urgent need for blood-based protein markers, and it could pave the way for further combined efforts to tackle the challenges posed by diseases with rare phenotypes.

# Materials and Methods

### Sample collection and study design

Serum and plasma samples were collected at four different clinical sites according to a collection protocol adopted within the framework of the BIO-NMD EU-FP7 program: Leiden University Medical Center in the Netherlands (LUMC), University of Ferrara in Italy (UNIFE)

**Figure 7.  Protein profiles of steroid-treated and non-treated muscular dystrophy patients and female carriers.**
Boxplots represent the five protein profiles across steroid-treated and non-treated DMD patients and female carriers. MFI values for MYL3, MDH2, CA3, ETFA, and TNNT3 in plasma and serum of steroid-treated, non-treated DMD patients and female carriers from UNEW are shown, where dark green, light green, and blue boxes represent treated DMD patients, non-treated DMD patients and female carriers, respectively. For each sample group, the box-and-whisker plot represents MFI values within lower and upper quantile (box), the median (horizontal line within box), percentiles of 5 and 95% (whiskers) and outliers (dots).

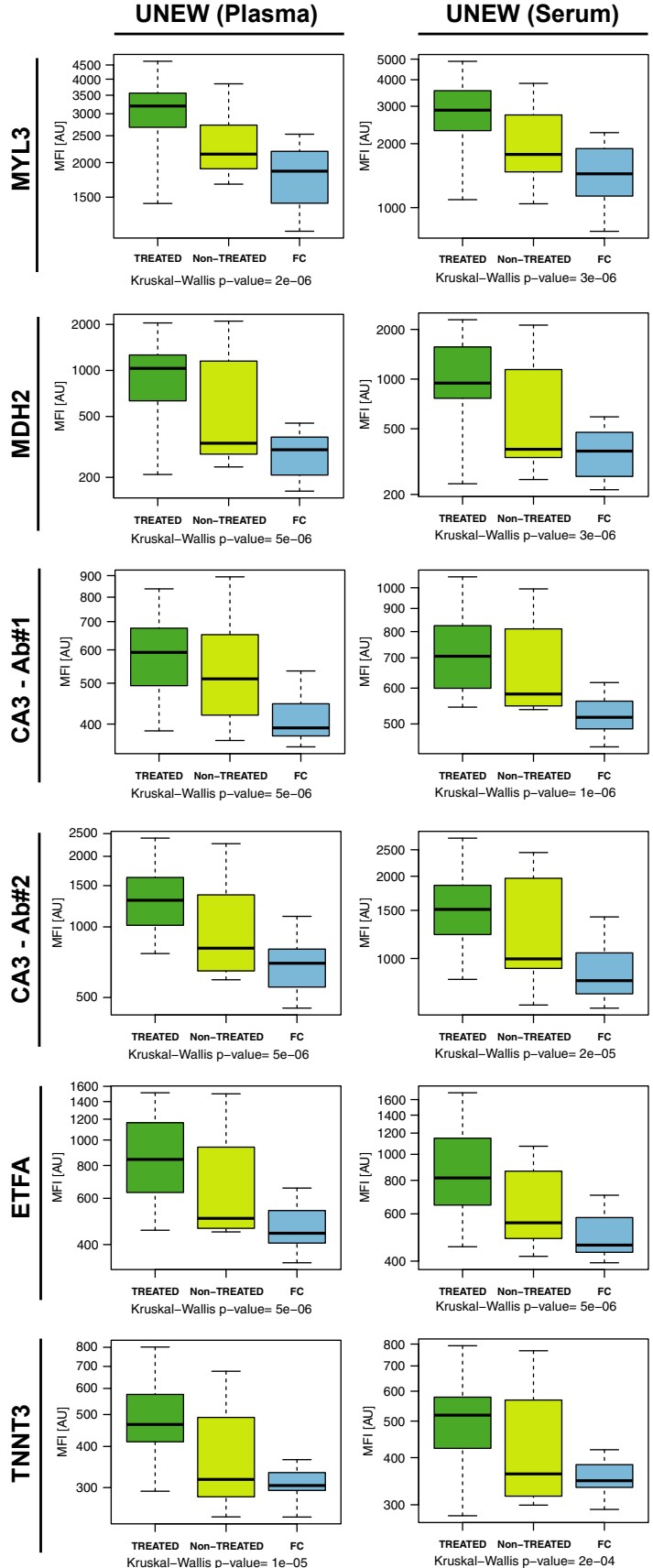

**Figure 7.**

and Newcastle University (UNEW) and University College of London (UCL) in United Kingdom. According to the standardized protocol established by the consortium, a total of 120 serum samples were collected at two of the sites and 225 plasma samples were collected at three of the sites (Table 1). Retrieval, storage, and use of samples were performed according to national policies regarding ethical treatment of human subjects. All cohorts included samples from patients with a genetically confirmed diagnosis of DMD but only the cohorts from UNIFE and UNEW included samples from individuals diagnosed with BMD. Controls were included in three cohorts: healthy individuals in the LUMC and UNIFE collections and female asymptomatic carriers in the UNEW collection. The sample collection from UNEW included matched serum and plasma samples from the same individuals. Together with the samples, information about gender, age, diagnosis, status of ambulation, and other relevant clinical parameters was assembled (Supplementary Dataset File S1). Collection of samples from patients and their use for research have been ethically approved by Ferrara Hospital Ethical Committee, Hammersmith Research Ethics committee, NRES Committee North East—Newcastle & North Tyneside 1 and LUMC Commissie Medische Ethiek and performed according to the principles set out in the WMA Declaration of Helsinki. Information about the samples was used as anonymized and aggregate data.

### Selection of candidate targets and design of the antibody array

A generous list of protein targets potentially associated with dystrophinopathies was assembled based on experimental evidence and/or theoretical analysis and annotated using the Pathway Studio software (Ariadne Genomic, Inc.) (Yuryev *et al*, 2006; Kotelnikova *et al*, 2012). The list included a total of 959 unique genes assembled based on 402 genes with experimental evidence for being associated with DMD, 248 genes expressed in healthy muscle, and 431 genes associated with muscular dystrophy as judged by literature search and Gene Ontology. For each gene, information about (i) experimental evidence for association with DMD, (ii) experimentally confirmed expression in normal skeletal muscle, (iii) involvement in pathways and cellular processes and/or Gene Ontology terms related to muscular dystrophy, and (iv) genes reported to be linked to DMD and other muscular dystrophies in the literature was compiled. The number of experimental evidence was scored and weighted to 40, 10, 2.5, and 10%, respectively. Characteristics of the corresponding gene products regarding secretion, detectability in serum and/or plasma, and evidence for expression in humans were also used to prioritize the genes. These characteristics were scored and weighted with 2.5, 5, and 2.5%, respectively. Each parameter was than multiplied by the given weight, and the weighted scores were summed. Furthermore, availability of antibodies toward these targets was checked in the Human Protein Atlas repository, and at least one antibody was selected for each target. The list was then supplemented with antibodies targeting known serum and plasma proteins, resulting in a set of 384 antibodies directed to 315 different proteins (Supplementary Dataset File S2). A detailed overview on number of unique protein targets and number of antibodies per target is provided in Supplementary Table S4. All antibodies were characterized and validated within the Human Protein Atlas framework on antigen microarrays, Western blots, and tissue

microarrays according to established protocols (Uhlén *et al*, 2010; Asplund *et al*, 2012).

### Generation of antibody suspension bead arrays

The concentration of all the antibodies was normalized using a liquid handling system (EVO150, TECAN) by diluting 1.6 μg of each antibody into 100 μl of 0.1 M 2-(N-morpholino)ethanesulfonic acid (MES) buffer (pH 4.5). Antibodies were then coupled to carboxylated, color-coded magnetic beads (MagPlex-C, Luminex Corp.) as per previously developed antibody-coupling protocols (Schwenk *et al*, 2008). In brief, $5 \times 10^5$ beads per bead identity were distributed across 96-well microtiter plates (Greiner BioOne), washed and re-suspended in phosphate buffer (0.1 M $NaH_2PO_4$, pH 6.2) using a plate washer (EL406, Biotek). Bead surfaces were activated by addition of 0.5 mg 1-ethyl-3(3-dimethylamino-propyl) carbodiimide (Pierce) and 0.5 mg N-hydroxysuccinimide (Pierce) in 100 μl phosphate buffer. After 20 min incubation on a shaker (Grant Bio), beads were washed with 0.1 M MES buffer. Pre-diluted antibodies were added to the beads using a liquid handler (SELMA, Cybio) and incubated for 2 h at RT. Three additional bead identities were functionalized either with 1.6 μg of rabbit IgG (Bethyl), 1.6 μg of in-house produced recombinant albumin binding protein, or without addition of any protein providing assay quality controls. Antibody-coupled beads were then washed 3× in PBS-T (1 × PBS, 0.05% Tween20), re-suspended in 50 μl of a storage buffer (Blocking Reagent for ELISA, Roche Applied Science) supplemented with 0.1% (v/v) ProClin (Sigma-Aldrich), and stored overnight at 4°C. A 384-plex antibody suspension bead array was prepared by combining equal volumes of each bead identity and followed by sonication for 3 min (Branson Ultrasonic Corp.). The bead array was stored at 4°C until further use. The coupling of each antibody on the beads was confirmed via R-Phycoerythrin-conjugated donkey anti-rabbit IgG antibody (Jackson ImmunoResearch) before performing the assay with patient samples.

### Pre-analytical preparation and labeling of plasma/serum samples

Neat serum/plasma samples were centrifuged for 10 min at 3,000 × *g* and aliquoted into microtiter plates with a liquid handling system (Freedom EVO150, TECAN). Three microliters of each sample was diluted in 22 μl of 1xPBS in new microtiter plates according to a plate layout design. This design allowed a randomized and balanced distribution of samples across multiple plates in terms of both categorical variables, namely sample origin (UNEW/ UNIFE/UCL/LUMC), blood preparation type (serum/plasma), and disease category (DMD/BMD/Control), and the quantitative variable of age tested by ANOVA test. The diluted and randomized samples were labeled utilizing biotin as previously described (Schwenk *et al*, 2008). Briefly, the diluted samples were incubated over 2 h at 4°C with a 10-fold molar excess of NHS-PEG4-Biotin (Pierce) calculated based on the assumption of an average molecular weight of 60 kDa and a plasma/serum total protein concentration of 60 mg/ml. The labeling reaction was quenched by addition of a 250-fold molar excess of 0.5 M Tris–HCl (pH 8.0) over biotin. After incubation with Tris–HCl for 20 min at 4°C, samples were stored back to −80°C until usage.

## Assay procedure and read-out

The biotin-labeled samples were diluted 1:50 using a liquid handler (SELMA, CyBio) in an assay buffer composed of 0.5% (w/v) polyvinylalcohol and 0.8% (w/v) polyvinylpyrrolidone (Sigma) in 0.1% (w/v) casein (Sigma-Aldrich) in PBS (PVXC) supplemented with 0.5 mg/ml rabbit IgG (Bethyl), yielding a total sample dilution of 1:500. Samples were then heat-treated at 56°C for 30 min and cooled to 20°C for 15 min in a thermo-cycler (Applied Biosystems). Then, 45 μl of heat-treated samples was added to 5 μl of the antibody suspension bead array distributed into a 384-well microtiter plate (Greiner BioOne). Subsequent to 16-h incubation on a shaker (Grant) at RT, beads were washed with 3 × 50 μl PBS-T using a plate washer (EL406, Biotek), followed by an incubation for 10 min with 50 μl of 0.4% paraformaldehyde in PBS-T. Beads were washed with 50 μl PBS-T and incubated with 50 μl of 0.5 μg/ml R-phycoerythrin labeled streptavidin (Invitrogen) in PBS-T for 20 min. Finally, beads were washed 3 × 50 μl PBS-T before addition of 60 μl of PBS-T for measurement in the FlexMap3D instrument (Luminex Corp.) utilizing the Luminex xPONENT software. At least 50 events per bead ID were counted, and binding events were displayed as median fluorescence intensity (MFI).

## Data analysis and statistics

All data analysis and visualizations were performed using R (Ihaka & Gentleman, 1996), unless otherwise indicated. MFI values were pre-processed, separately for serum and plasma, using probabilistic quotient normalization (PQN) (Dieterle *et al*, 2006) accounting for any potential sample dilution effects (Kato *et al*, 2011), and the PQN-normalized data were used in the statistical analyses displayed in the figures and tables. Principal component analysis (PCA) was performed to confirm that there was no systematic variation in the dataset driven by sample origin or assay plate and that there were no outlier samples. Heatmaps with unsupervised hierarchical clustering were generated using the Qlucore Omics Explorer software v2.3 (Qlucore) for exploratory analysis, which utilizes *t*-test for two-group comparisons and F-test for multi-group comparisons as an in-built statistical filtering functionality. Self-organizing tree algorithm (SOTA) was applied using the R package 'cIValid' (Herrero *et al*, 2001) on scaled and centered MFI values across DMD, BMD, and CONT/FC groups for an unsupervised and divisive clustering of protein profiles across DMD, BMD, and CONT/FC groups. All correlation coefficients were calculated using nonparametric Spearman's correlation.

The nonparametric Wilcoxon rank-sum and Kruskal–Wallis tests were applied to PQN-normalized and log2-transformed data to calculate *P* values. Differences in protein profiles between compared groups were denoted statistically significant if they concordantly revealed *P* values < 0.01 in different cohorts and blood preparation types, without multiple testing correction. The intersection of proteins revealing *P* values < 0.01 was identified by using the R package 'VennDiagram' and visualized with Venn diagrams (Chen & Boutros, 2011). Datasets including only these intersecting proteins were analyzed using the web-based tool 'PanelComposer' (Jeong *et al*, 2012) employing logistic regression to compare the classification power of single or different combinations

### The paper explained

#### Problem
Young boys affected by Duchenne muscular dystrophy, a rare and severe genetic disease, are diagnosed using an array of different genetic, enzymatic, histopathologic and physical tests. These tests correlate poorly with disease severity and are often affected by other factors than the disease such as the age, the overall well-being, the level of understanding, and the ability of the patient to cooperate with the clinicians. As the disease progresses, patients lose their ability to walk and stand, making physical tests more difficult and painful to perform. Another test often used is analysis of muscle tissue biopsies collected through invasive procedures causing great discomfort to the patients. New approaches and tests are required to improve clinical management of muscular dystrophies and accurately determine the severity of the disease, disease progression, and treatment outcome.

#### Results
Protein levels in blood show promise for providing clinically relevant information to monitor patient health status. Comparing levels of proteins in blood, we identified protein profiles that discriminate between patients affected by Duchenne muscular dystrophy with different degrees of severity. Four proteins, carbonic anhydrase III (CA3), myosin light chain 3 (MYL3), malate dehydrogenase 2 (MDH2), and electron transfer flavoprotein A (ETFA), are more abundant in blood from patients with DMD in comparison with healthy individuals. The results obtained are in agreement in samples collected from four clinical sites and concordant in both serum and plasma.
The last two proteins also correlate with the patient ambulation status and respiratory insufficiency in different subgroups of patients.

#### Impact
In the context of muscular dystrophies, there is a need for molecular biomarkers that can be used to determine disease severity and to monitor disease progression over time. The identified protein markers are easily accessible and provide information that can improve preventive clinical management of the disease and selection of individualized treatment regimes. Given the short life expectancy of the patients with Duchenne muscle dystrophy, development of more accurate tests to improve clinical management of the disease will have a great impact on patient life quality. Furthermore, this study represents an important example for how more insights into proteins altered in the circulation of patients with rare diseases can be effectively studied across various clinical sites using affinity-proteomics approaches.

of proteins, where leave-one-out method was selected as cross-validation option. Multivariate binary logistic regression was performed for protein panels suggested by PanelComposer, and the respective ROC curves were generated using the R package 'EpiCalc'. For each comparison, proteins with *P* values < 0.01 and confirmed by analysis of all relevant cohorts were included in the test panels. The data have been deposited in the ArrayExpress database with the accession number E-MTAB-2564.

**Supplementary information** for this article is available online: http://embomolmed.embopress.org

## Acknowledgements
We thank Mun-Gwan Hong for his help with the plate layout design script and the whole group of Biobank Profiling-Affinity Proteomics at SciLifeLab Stockholm. We also acknowledge the entire staff of the Human Protein Atlas

for their efforts in generating the antibodies, the clinical teams, and biobank staff at Newcastle, London and the NHMB Telethon Italy biobank and the Italian Duchenne Parent Project for the DMD patients' database setup and helpful collaboration. We would especially like to thank all the patients who donated samples for this study. This study was supported by the European Community Framework Programme 7 grant agreement no. 241665 for the BIO-NMD project; the ProNova VINN Excellence Centre for Protein Technology (VINNOVA, Swedish Governmental Agency for Innovation Systems); the Knut and Alice Wallenberg Foundation; SciLifeLab Stockholm and Great Ormond Street Children's Charity. The MRC Center for Neuromuscular Diseases Biobank in London and Newcastle is supported by the Medical Research Council (UK) and is a partner of EuroBioBank (www.eurobiobank.org). The study was also supported by TREAT-NMD (EU FP6 contract no. 036825) and the TREAT-NMD Alliance (www.treat-nmd.eu).

## Author contributions

BA, CAS, PN designed the hypothesis, the experimental setup, and together with AC and HL designed the study. AF, FM, VS, KB, and EN were involved with the ethical conduct of the study, supported clinical stratification of patients, and coordinated sample collection at their sites. HL, AC, SC, EB, LP, and MH recruited patients, collected samples, and acquired clinical data. ES, YLP, AF, SC have suggested targets and ES and YLP have designed the strategy for selection and prioritization of targets. BA performed the experimental work and together with JMS the data analysis. BA, CAS, PN, AF, HL, FM, PACtH, AAR, PS interpreted the results. FP contributed with immunohistochemical staining of tissues and interpretation of results. BA, CAS, PN, JMS, AC, and HL wrote the manuscript. AF, FM, MU, PACtH, AAR, VS, HL, CAS, ES, KB, and YLP contributed to the original grant application.

## Conflict of interest

The authors declare that they have no conflict of interest.

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
