## [Review Process File · EMBO Molecular Medicine]

Affinity proteomics within rare diseases: A BIO-NMD study for blood biomarkers of muscular dystrophies

Burcu Ayoglu, Amina Chaouch, Hanns Lochmüller, Luisa Politano, Enrico Bertini, Pietro Spitali, Monika Hiller, Eric Niks, Francesca Gualandi, Fredrik Pontén, Kate Bushby, Annemieke Aartsma-Rus, Elena Schwartz, Yannick Le Priol, Volker Straub, Mathias Uhlén, Sebahattin Cirak, Peter A.C. 't Hoen, Francesco Muntoni, Alessandra Ferlini, Jochen M. Schwenk, Peter Nilsson, Cristina Al-Khalili Szigyarto

Corresponding author: Cristina Al-Khalili Szigyarto, KTH-Royal Institute of Technology

Review timeline:

Submission date:	27 November 2013
Editorial Decision:	23 December 2013
Appeal:	10 January 2014
Editorial Decision:	13 January 2014
Revision received:	29 March 2014
Editorial Decision:	29 April 2014
Accepted:	15 May 2014

Transaction Report:

Editor: Céline Carret

1st Editorial Decision

23 December 2013

Thank you for the submission of your manuscript "Profiling geographically dispersed cohorts reveals protein markers in blood for Duchenne muscular dystrophies". We have now heard back from the three referees whom we asked to evaluate your manuscript.

As you will see, the referees acknowledge the potential importance of the study and the quality of the data provided. However, all three clearly highlight that this interest and importance are mainly relevant to the immediate community. As EMBO Molecular Medicine aims at publishing research articles of general interest for a broad readership, I am afraid that I see no other choice than returning the manuscript to you at this stage with the message that we can not offer to publish it.

I am really sorry that I could not bring better news sooner, but hope that the referees' comments will be helpful in identifying a more suitable venue for your article.

***** Reviewer's comments *****

Referee #1 (Remarks):

Paper description: The authors (all 23 of them) describe a significant Duchenne's Muscular Dystrophy blood proteomics study on 245 people with DMD (130), Becker's Muscular Dystrophy

(33), female carriers (16), and controls (66). DMD is caused by null mutations in the dystrophin gene (which is encoded on the X chromosome, and thus is a fatal disease of boys and young men), while Becker's also is caused by less severe mutations in dystrophin.

Because both serum and plasma were used, a total of 345 samples were analyzed. The samples were obtained from four clinical sites

1. UNEW - University of Newcastle (100 individuals)
2. LUMC - Leiden University Medical Center (20 individuals)
3. UCL - University College London (40 individuals)
4. UNIFE - University of Ferrara (85)

The authors analyzed the concentrations of 315 unique human proteins using an assay they have developed and described previously, and which I describe here in short form; for the work in the paper 384 antibodies were used. The assay consists of chemically affixing each antibody to a Luminex color-coded magnetic bead. The samples themselves - neat plasma or serum - were clarified by centrifugation, after which 3 ml of each human sample was added to 22 ml of PBS, biotinylated with a 10-fold molar excess of NHS-PEG4-Biotin for two hours at 4 °C, processed and refrozen until used in the assay. When assayed, the frozen human samples are retrieved and diluted 50X into the assay buffer (for a total sample dilution of 500X), incubated at 56 °C for 30 minutes (to partially denature the biotinylated proteins in the sample because the antibodies were prepared against peptide antigens), and then added to the Luminex beads and incubated for 16 hours, during which time the antibody-conjugated beads capture the serum or plasma proteins. After washing three times, the antibody-antigen complexes are cross-linked by treatment with 0.4% paraformaldehyde and labeled/stained with phycoerythrin-labeled streptavidin (which binds to the biotin on the target proteins) for quantification on the Luminex FlexMap3D instrument. The reported CVs for the assay are 8%-9%.

Although the Luminex platform is often used for ELISA assays, the platform used for this work does not utilize antibody sandwiches - rather, the mono-specific polyclonal rabbit antibodies developed by Uhlen's lab are specific enough to be used as the only specificity determinant for human protein capture. The assay almost looks like a sandwich, but it is not: the phycoerythrin-labeled streptavidin is just a stain for the biotinylated human antigens.

The data were then analyzed by a variety of methods, and biomarkers were found with p-values of < 0.01 and a few with p-values of < 0.001 (Table 2). Twelve human proteins were identified as potential biomarkers for these medical conditions, largely thought to be proteins expressed in muscle and which are elevated in the blood of patients because of muscle cell death and protein leakage into blood.

Positive comments: The authors strongly and convincingly make the case for the value of protein biomarkers for patient management for the dystrophinopathies - the authors argue that biomarkers would be used to monitor patient health and would become valuable for patient stratification during clinical trials of new drugs (of which there are several undergoing clinical development). While the findings are not extraordinarily unusual (much of the DMD biomarker literature, starting with blood levels for creatine kinase, is aimed at dying muscle cells and the leakage of muscle proteins into blood), one gets the sense that this manuscript sets a new standard for the search for DMD biomarkers. As such, the paper should be published as an aide to the DMD community (and see below).

Improvements to the paper: I had several suggestions to make the paper even more important for the DMD community. I tried to suggest improvements that could be achieved by writing rather than by extensive new experiments.

1. I would like the authors to extend the discussion of "Study design", particularly the section about target selection which is handled in one short paragraph. The Uhlen lab has prepared antibodies to ~10,000 (or more) human proteins, and yet the authors limit the analysis to 315 human protein targets (using 384 antibodies). The results obtained must be a function of that selection. I would like the authors to tell the readers why they analyzed only 315 targets and why this particular set. In my view this is the heart of the paper, and the authors have chosen a particular lamppost under which to look; the readers need insight into those choices. I also would like the authors to tell us how many targets they could have analyzed had they chosen to expand the list - that is, how large a study could have been performed on their platform.
2. For the 69 "extra" antibodies (384-315=69) in the analysis I would like the authors to compare the data for each pair so that the reader is led to conclude that duplicates are not required because the 69 duplicates each gave more or less the same answer.
3. I would like the authors to tell the readers what number of the 384 measurements made on each sample did not rise above the counts expected in a heterologous sample (such as yeast extract, for

example) - that is, of the 384 measurements, how many were not significantly signaling.

4. In a related suggestion, I would like the authors to comment on those proteins in the list of 315 that they might have expected to perform like the biomarkers they did find. Specifically, I am wondering if many muscle proteins that the authors expected to leak into blood during cell death were not elevated in the DMD patients, and, if that is the case, I would like the authors to rationalize those data.

5. Finally, I would like the authors to comment on the likely abundance of the proteins that are biomarkers in the study and on assay specificity (the authors state that at least 50 events were counted on the Luminex assay - I am asking about the background in the assay). This comes from thinking about the assay itself, which was performed on a 500X dilution of the plasma or serum samples - that is, performed on 0.2% sample rather than 100% or 50% serum or plasma which might have enabled less abundant proteins to be counted. The Uhlen lab has studied their antibodies exhaustively, and they have a clear idea about peptide cross-reactivity; is this in part the reason for the use of 0.2% sample?

Recommendation: I would accept this paper for publication and I would ask the authors to discuss the five points I listed above. The paper is important for an underserved community of patients and families.

Referee #2 (Comments on Novelty/Model System):

Manuscript EMM-2013-03724 is a well executed study that has established several serum-associated muscle proteins as potential biomarkers for relatively rare genetic disorders, the group of X-linked muscular dystrophies.

Referee #2 (Remarks):

Manuscript EMM-2013-03724 by Cristina Al-Khalili Azigyarto and co-workers entitled 'Profiling geographically dispersed cohorts reveals protein markers in blood for Duchenne Muscular dystrophies' describes a well-executed screening study of a large and diverse cohort of blood samples from patients afflicted with X-linked muscular dystrophies. The antibody-based approach is well designed and properly presented, and appears to be a highly suitable way to discover novel protein biomarker candidates in serum samples. Since there is an urgent need for reliable biomarkers to improve diagnosis and prognosis of dystrophic patients, as well as determine the value of experimental treatments, this study is an important contribution to the field of muscular dystrophy research.

To clarify a few points in this manuscript, the following issues should be addressed by the authors:

(i) One of the key findings of the antibody-based proteomic screening exercise presented in this paper is the elevated concentration of the CA3 isoform of carbonic anhydrase, which is a relatively common and abundant protein associated with muscle tissue that also appears to exhibit a fibre type-specific distribution. The authors are encouraged to better outline these points in their discussion with respect to:

- Potential muscular dystrophy-unrelated effects: Is it possible that CA3 levels in blood samples are majorly affected by certain crush injuries, mild forms of exercise, a variety of drug effects or pathologies unrelated to muscular dystrophy? It is certainly well known that both exercise or injury trigger elevated serum CA3 levels. Please see: Brancaccio P, Lippi G, Maffulli N. Biochemical markers of muscular damage. *Clin Chem Lab Med.* 2010 Jun;48(6):757-67.

- Fibre type specific distribution of CA3: A previous immunocytochemical and biochemical study suggests that cytosolic carbonic anhydrase isoform CA3 is present at higher levels in soleus muscle, as compared to deep vastus lateralis muscle and superficial vastus lateralis muscle, which are composed of predominantly type I, IIa, and IIb fibers, respectively. Please see: FrÈmont P, Charest PM, CÙtÈ C, Rogers PA. Carbonic anhydrase III in skeletal muscle fibers: an immunocytochemical and biochemical study. *J Histochem Cytochem.* 1988 Jul;36(7):775-82. In addition, chronic electrostimulation-induced transitions from fast to slower contracting muscle phenotypes is associated with marked increase in CA3 mRNA levels, agreeing with the concept of a fibre type-

specific distribution of this enzyme in skeletal muscle. Please see: Brownson C, Isenberg H, Brown W, Salmons S, Edwards Y. Changes in skeletal muscle gene transcription induced by chronic stimulation. *Muscle Nerve*. 1988 Nov;11(11):1183-9. This issue of the fibre type distribution of the novel biomarker candidate CA3 should be outlined in more detail and it should be discussed whether this fact may have an effect on the pathobiochemical leakage of this protein in muscular dystrophy.

- Additional reference on CA3 suitability: Besides the already quoted references on CA3, a previous paper by Vönnänen and co-workers has studied CA3 leakage as a potential biomarker of type I skeletal muscle fibres in polymyositis, muscular dystrophies, amyotrophic lateral sclerosis and other neurogenic diseases. Please see: Vönnänen HK, Takala TE, Tolonen U, Vuori J, Myllylä VV. Muscle-specific carbonic anhydrase III is a more sensitive marker of muscle damage than creatine kinase in neuromuscular disorders. *Arch Neurol*. 1988 Nov;45(11):1254-6.

(ii) Standardization of sampling procedure: Analytical studies involving blood, serum or plasma samples involve crucial steps of specimen handling, fractionation and storage. Do the authors have an indication how critical the choice of tubes, material, chemicals, centrifugation steps, handling time and storage method is for the production of highly reproducible findings? It is known from large-scale proteomic studies of serum and plasma samples from human blood that even small changes in the protocol can have substantial effects on the measurement of protein markers. In this respect, would the implementation of the suggested new serum biomarkers require a common international protocol independent of national guidelines to establish a proper screening system for patients with dystrophinopathies.

(iii) The authors are encouraged to discuss in more detail the actual robustness of their newly suggested assay system, detailing the best usage of biomarker panels, the most suitable protein marker signature for differential analyses of dystrophinopathies, and the critical issue of sensitivity versus specificity.

(iv) Although this study represents a large-scale proteomic style study that is based on high-throughput antibody technology, the paper does not mention or discuss in much detail the large volume of previous proteomic studies, that certainly have a certain degree of relevance to this screening exercise. For a recent comprehensive review of the proteomic analysis of muscle specimens, please see: Holland et al. Proteomics of the dystrophin-glycoprotein complex and dystrophinopathy. *Curr Protein Pept Sci*. 2013 Oct 4. [Epub ahead of print] PMID: 24106963. This review also contains a discussion on the very recent publication of the muscle secretome from mdx muscle cells by Partridge and co-workers, which might be of interest to the general discussion on leakage versus secretion of distinct protein populations in pathological muscles. Please see: Duguez S, Duddy W, Johnston H, Lainé J, Le Bihan MC, Brown KJ, Bigot A, Hathout Y, Butler-Browne G, Partridge T. Dystrophin deficiency leads to disturbance of LAMP1-vesicle-associated protein secretion. *Cell Mol Life Sci*. 2013 Jun;70(12):2159-74.

(v) Potentially relevant references on the proteomic analysis of mdx plasma: Two previous studies have focused on the evaluation of potential changes in the plasma from the dystrophic mdx animal model of Duchenne muscular dystrophy. The authors should evaluate whether any findings of these studies correlate to their analysis of human blood samples and whether fundamental differences exist between the mild phenotype of the mdx mouse and the highly progressive form of human DMD. Please see: Alagaratnam, S.; Mertens, B.J.; Dalebout, J.C.; Deelder, A.M.; van Ommen, G.J.; den Dunnen, J.T.; 't Hoen, P.A. Serum protein profiling in mice: identification of Factor XIIIa as a potential biomarker for muscular dystrophy. *Proteomics*, 2008, 8, 1552-1563; and Colussi, C.; Banfi, C.; Brioschi, M.; Tremoli, E.; Straino, S.; Spallotta, F.; Mai, A.; Rotili, D.; Capogrossi, M.C.; Gaetano, C. Proteomic profile of differentially expressed plasma proteins from dystrophic mice and following suberoylanilide hydroxamic acid treatment. *Proteomics Clin. Appl.*, 2010, 4, 71-83.

(vi) Human muscle biopsies: If sufficient numbers of patient muscle samples are available, potential changes in the novel markers could be evaluated on the level of muscle tissue in order to possibly determine whether distinct differences exist between disease-associated sarcolemmal leakage and natural cellular secretion processes. Confocal microscopy would only require very small samples, if available, to determine potential changes in the subcellular location and/or overall concentration of the new markers within fibres.

(vii) Minor point: In the References list, 5 references are missing volume or page numbers. This includes the References: Fagerberg et al..., Mercuri and Muntoni..., Moat et al..., Nishita et al..., and Staunton et al...!

Referee #3 (Remarks):

This paper contains a lot of data on the profiling of proteins expressed in 345 blood samples from DMD patients collected from 4 different clinical centres. The data show significant differences in four proteins between patients and controls which correlate with the age in sub cohorts of patients and with the clinical severity as evidenced from the data on BMD patients.

This is a disease where biomarkers are badly needed to monitor the progress of patients in clinical trials and these data would need to be validated by other groups. As far as I know, Somalogics using a different technology are the only other investigators to study this as extensively.

The data are of high quality and interesting. They are more valuable than most data sets because the patients are collected from four clinical centres.

It is difficult to predict whether these data will be confirmed by others. The authors relate their observations to plausible mechanisms but I might have expected to observe more than one protein derived from the same pathway of the disease process. Certainly network analysis would add more significance to the data.

These results are important. However, since the four protein markers are independent markers I think further work is needed to support their validity as biomarkers in DMD.

Appeal

10 January 2014

Thank you very much for your reply and the brief comment about our manuscript (EMM-2013-03724).

We highly appreciated the reviewers comments and the relevant criticism that was raised. We found the overall impression from the referees positive and we are well prepared to address these accordingly, thus the rejection came to us as a surprise. From our perspective and based on our reading of the referees' comments, we are not yet sure what scientific reasons have motivated not to allow a revised manuscript to enter the next stage of the review process.

Of course, one important aspect raised is the scope of the manuscript, which in its present form can be considered narrow even though it addresses an important aspect for many rare diseases. We again are certain to be able to address this in a revision. We therefore kindly ask you to reconsider your decision and offer the possibility to revise our work so it better suits the journal.

Together with my co-authors we can at this point complement the manuscript with information that widens the scope of the manuscript to be more appealing to the EMBO MolMed readers. We can cover all points raised by Referee 1 and 3 and majority of the comments raised by Referee 2. We would also like to emphasize that our work has been submitted to EMBO Molecular Medicine in particular to reach an audience beyond the affected community, their families, physicians and scientists involved in the muscular dystrophy research field.

Our approach has proven that by using four different cohorts of small size, potential biomarkers are identified and can be validated if concordant results are obtained in both serum and plasma. This has a great importance for research on all rare disorders where sample availability is a limiting aspect and an impediment for the research field. In comparison to biomarker discovery for disorders that affect a large number of patients, our approach shows the possibility of moving the boundaries of

knowledge forward even when only limited number and volumes of samples are available if a rigorous statistical analysis is applied. This study represents an important example for how more insights into proteins altered in the circulation of patients with rare diseases can be studied across various clinical sites.

Given the limitations of current proteomic methods and the continuous need to enhance performance, the chosen approach is particularly useful for body fluids, which are more patient friendly to obtain. In line with this efforts we have performed our analysis and identified protein markers from minute amounts of both serum and plasma. The value of our study for this research field is in identification of biomarkers that are intended to be robust and not affected by covariates or the differential preparation of samples.

The reported presence of muscle proteins in serum and plasma of patients affected by muscle dystrophies, implicitly proves that tissue leakage can mirror clinical parameters and confirms that muscle deterioration can influence the blood composition. Muscle deterioration occurs at a rapid rate in patients affected by 30 different muscle dystrophies but also at slower rate in healthy aging individuals. Our work thus opens the possibility to investigate blood markers for each and one of these conditions.

2nd Editorial Decision

13 January 2014

Thank you for your letter asking us to reconsider our decision on your article.

After discussing within the team, including our chief editor, re-reading the referees' comments and your appeal letter, we have decided to invite you to revise your article. However, I would like to strongly encourage you to broaden your study as you suggested in order to make it more appealing to our general readership. I would also like to suggest changing the title to something more impactful. Obviously, the referees' concerns have to be addressed experimentally when appropriate and a detailed response to their comments must be provided along with additional explanations, references incorporation and discussion.

I look forward to seeing a revised form of your manuscript as soon as possible.

1st Revision - authors' response

29 March 2014

Referee #1 (Remarks):

Paper description: The authors (all 23 of them) describe a significant Duchenne's Muscular Dystrophy blood proteomics study on 245 people with DMD (130), Becker's Muscular Dystrophy (33), female carriers (16), and controls (66). DMD is caused by null mutations in the dystrophin gene (which is encoded on the X chromosome, and thus is a fatal disease of boys and young men), while Becker's also is caused by less severe mutations in dystrophin. Because both serum and plasma were used, a total of 345 samples were analysed. The samples were obtained from four clinical sites

1. UNEW - University of Newcastle (100 individuals)
2. LUMC - Leiden University Medical Centre (20 individuals)
3. UCL - University College London (40 individuals)
4. UNIFE - University of Ferrara (85)

The authors analysed the concentrations of 315 unique human proteins using an assay they have developed and described previously, and which I describe here in short form; for the work in the paper 384 antibodies were used. The assay consists of chemically affixing each antibody to a Luminex color-coded magnetic bead. The samples themselves - neat plasma or serum - were clarified by centrifugation, after which 3 ml of each human sample was added to 22 ml of PBS,

biotinylated with a 10-fold molar excess of NHS-PEG4-Biotin for two hours at 4C, processed and refrozen until used in the assay. When assayed, the frozen human samples are retrieved and diluted 50X into the assay buffer (for a total sample dilution of 500X), incubated at 56oC for 30 minutes (to partially denature the biotinylated proteins in the sample because the antibodies were prepared against peptide antigens), and then added to the Luminex beads and incubated for 16 hours, during which time the antibody-conjugated beads capture the serum or plasma proteins. After washing three times, the antibody-antigen complexes are cross-linked by treatment with 0.4% paraformaldehyde and labelled/stained with phycoerythrin-labelled streptavidin (which binds to the biotin on the target proteins) for quantification on the Luminex FlexMap3D instrument. The reported CVs for the assay are 8%-9%.

Although the Luminex platform is often used for ELISA assays, the platform used for this work does not utilize antibody sandwiches - rather, the mono-specific polyclonal rabbit antibodies developed by Uhlen's lab are specific enough to be used as the only specificity determinant for human protein capture. The assay almost looks like a sandwich, but it is not: the phycoerythrin-labelled streptavidin is just a stain for the biotinylated human antigens.

The data were then analysed by a variety of methods, and biomarkers were found with p-values of < 0.01 and a few with p-values of < 0.001 (Table 2). Twelve human proteins were identified as potential biomarkers for these medical conditions, largely thought to be proteins expressed in muscle and which are elevated in the blood of patients because of muscle cell death and protein leakage into blood.

Positive comments:

The authors strongly and convincingly make the case for the value of protein biomarkers for patient management for the dystrophinopathies - the authors argue that biomarkers would be used to monitor patient health and would become valuable for patient stratification during clinical trials of new drugs (of which there are several undergoing clinical development). While the findings are not extraordinarily unusual (much of the DMD biomarker literature, starting with blood levels for creatine kinase, is aimed at dying muscle cells and the leakage of muscle proteins into blood), one gets the sense that this manuscript sets a new standard for the search for DMD biomarkers. As such, the paper should be published as an aide to the DMD community (and see below). “

Author's comment:

We would like to thank the referee for the positive comments regarding our manuscript. Proteomics methods, including array-based protein profiling approaches, are being increasingly used for identification of biomarkers especially in cancer research but very seldom, if not at all, in the context of rare, genetic diseases. Therefore, one of our main aims in this study and within the BIO-NMD consortium was to apply and demonstrate the potential of an affinity-proteomics approach in the quest for plasma protein markers in a rare and genetic disease. As also highlighted by the referee, the approach we presented in this study will hopefully stimulate similar efforts utilizing proteomics approaches to explore the effects of genetic defects in rare diseases on the protein level in blood-derived samples.

Continuation Referee #1 (Remarks):

I had several suggestions to make the paper even more important for the DMD community. I tried to suggest improvements that could be achieved by writing rather than by extensive new experiments.

1. I would like the authors to extend the discussion of "Study design", particularly the section about target selection, which is handled in one short paragraph. The Uhlen lab has prepared antibodies to ~10,000 (or more) human proteins, and yet the authors limit the analysis to 315 human protein targets (using 384 antibodies). The results obtained must be a function of that selection. I would like the authors to tell the readers why they analysed only 315 targets and why this particular set. In my view this is the heart of the paper, and the authors have chosen a particular lamppost under which to look; the readers need insight into those choices.

Author's response:

We agree on the need to clarify and provide more insight into the target selection. The key aspect for selection of targets was the hypothesis that muscle creatine kinase leakage into the blood stream is accompanied by other proteins involved in muscle function. In line with this, proteins were selected and prioritized regarding potential association with muscle function, mainly from knowledge generated by different partners in the BIO-NMD consortium as well as from a generous and inclusive literature search. In addition to these biological considerations, we have checked for the availability of quality-controlled antibodies towards these targets within the Human Protein Atlas and applied a threshold for antibody concentration that would allow for a dilution of antibody volumes using a robotic liquid handling device. Examining both these various biological factors and practicalities revealed a set of 380 available antibodies (excluding 4 assay controls) targeting 315 unique protein targets. Please also refer to our reply regarding the next comment where we provided an explanation for the rationale behind creating the bead arrays consisting of up to 384 different antibodies.

Author's improvements to the manuscript:

Selection of targets and antibodies has now been described in more detail in the sub-section of "Selection of candidate targets and design of the antibody array" (under Materials & Methods, page 21-22) and in the sub-section of "Study and experimental design" (under Results, page 5-6).

Continuation Referee #1 (Remarks):

I also would like the authors to tell us how many targets they could have analysed had they chosen to expand the list - that is, how large a study could have been performed on their platform.

Author's response:

The dimension of the study is determined on one hand by the number of available, suitable antibodies and on the other hand by the degree of multiplexing possible per single round analysis. We utilize bead arrays consisting of up to 384 different antibody-coupled color-coded bead identities and the read-out system allows analysis of up to 384 samples per run in a 384-well microtiter plate format. Since the consumed sample volume per 384 antibodies is small ($< 1 \mu\text{l}$), 50 μl of crude samples would theoretically suffice to employ binding reagents against all human proteins in about 50 rounds of analysis. The biggest challenge for affinity proteomics studies is though the availability of stringently validated antibodies to expand studies to cover larger fractions of the human proteome (Stoevesandt & Taussig, 2012; Colwill *et al*, 2011). There are though initiatives, such as the Human Protein Atlas (HPA) addressing this challenge, where more than 42,000 polyclonal antibodies have so far been validated and approximately 21,000 of them (representing 16,000 unique proteins) are presented at www.proteinatlas.org. Therefore, in theory the target list could have been extended. We have in other projects made very large efforts and profiled 10,000 antibodies on hundreds of samples. However, when the antibody set is built in an entirely hypothesis-free manner, we know that the fraction of proteins being clearly detected in plasma is relatively low (Schwenk *et al.*, unpublished data), which is expected with the current sensitivity determined to be in the low ng/ml range. Therefore, the experimental design here was chosen both to focus on a hypothesis-driven set of protein targets with known or predicted relation to muscular dystrophies and to still allow multiplexed profiling of large number of proteins.

Continuation Referee #1 (Remarks):

2. For the 69 "extra" antibodies (384-315=69) in the analysis I would like the authors to compare the data for each pair so that the reader is led to conclude that duplicates are not required because the 69 duplicates each gave more or less the same answer.

Author's response:

As the referee highlights, for 56 (out of 315) target proteins our antibody set contained two or more antibodies raised against different parts of the target protein. A more detailed overview of this is provided now in **Supporting Information Table 4**.

These antibodies are all polyclonal antibodies and thereby also the antibody pairs by definition are not “duplicates” and identical. Furthermore, these antibody pairs have been raised against different parts of their respective target protein and in many cases they even target different splice variants of the target protein. The screening is based on a single binder assay format and the main reason to include more antibodies per target is basically to increase the coverage for some of the prioritized proteins. There are several examples in the herein presented dataset, where the paired antibodies are strongly correlating and thereby validate each other, such as for CA3 and MDH2.

We have investigated the correlation between protein profiles generated by these 75 antibody pairs, separately in all plasma (n=225) and serum (n=136) samples. Despite the fact that not all the targets are necessarily detectable and that the majority of the antibody pairs target different epitopes/isoforms with different accessibility, 16 pairs still revealed a Spearman’s Rho ≥ 0.4 either in serum or plasma, with 6 of these pairs being common for both blood preparation types. The 6 targets for which we observed well-correlating protein profiles both in serum and plasma included one of our highlighted candidates, namely CA3 (Spearman’s Rho in serum=0.83, in plasma=0.80), whereas the antibody pairs targeting MDH2 revealed correlating protein profiles only in plasma (Spearman’s Rho=0.44) but not in serum (**Supporting Information Figure 3**).

Author’s improvements to the manuscript:

Regarding this point, we have now added **Supporting Information Table 4** and **Supporting Information Figure 3** and referred to the related figure in the sub-section of “Study and experimental design” (under Results, *page 6*) and in the sub-section of “Protein profiles associated with muscular dystrophy” (under Results, *page 8*).

Continuation Referee #1 (Remarks):

3. I would like the authors to tell the readers what number of the 384 measurements made on each sample did not rise above the counts expected in a heterologous sample (such as yeast extract, for example) - that is, of the 384 measurements, how many were not significantly signalling.

Author’s response:

Our understanding of this question is that the reviewer is interested in the number of antibodies that did not reveal a signal above noise level. As the referee suggests, one option of addressing this question could be parallel analysis of *e.g.* a yeast extract sample, although the presence of a noticeable number of evolutionarily conserved proteins as well as a new analytical context (sample environment and composition, effect of lysis on extracted proteins *etc.*) needs to be considered and thoroughly evaluated. We generally address this within each assay by including beads prepared with a protein-free solution and with normal rabbit IgG (without a defined specificity). Signal intensities revealed by these two bead identities indicate if antibodies have captured proteins beyond the non-specific background binding. The spread of non-specific binding across all samples is then used to determine the “noise” level for a given assay. The boxplots in **Supporting Information Figure 2** represent the signal intensities over all serum or plasma samples for each antibody, where the upper endpoints of the 99% confidence intervals for the negative control beads coupled either to no antibody or to non-specific rabbit IgG are shown. For convenience, we have now summarized in **Supporting Information Table 1** the number and percentage of antibodies revealing signal intensities below these thresholds marked in **Supporting Information Figure 2**. The union of these sets of antibodies included a total of 69 different antibodies. We can therefore conclude 69/382, namely 18% of the antibodies included in our antibody selection revealed signal intensities at noise level.

Author’s improvements to the manuscript:

Regarding this point, we have now added **Supporting Information Table 1** and referred to the related table and figure in the sub-section of “Study and experimental design” (under Results, page 6).

Continuation Referee #1 (Remarks):

4. In a related suggestion, I would like the authors to comment on those proteins in the list of 315 that they might have expected to perform like the biomarkers they did find. Specifically, I am wondering if many muscle proteins that the authors expected to leak into blood during cell death were not elevated in the DMD patients, and, if that is the case, I would like the authors to rationalize those data.

Author’s response:

Out of the 315 protein targets in our study, 112 of them were included due to positive immunohistochemical staining in muscle tissue based on the data generated within the framework of Human Protein Atlas. These 112 proteins shown to be expressed in muscle tissue were targeted in our assay by 153 antibodies. We have now performed an unsupervised cluster analysis using the scaled and centred median MFI values across DMD, BMD and CONT/FC groups within UNEW and UNIFE cohorts for each of these 153 protein profiles using the self-organizing tree algorithm (SOTA). As shown in the new **Supporting Information Figure 8**, this analysis revealed which of these protein profiles shows a “DMD increased” and/or “BMD increased” trend as compared to the control groups: In UNIFE cohort there were a total of 65 proteins (targeted by 73 antibodies) and in UNEW plasma and serum cohorts there were a total of 62 proteins (targeted by 74 and 73 antibodies, respectively). The union of these 3 sets of “DMD/BMD increased” proteins consists of 94 (out of 112) targets and the intersection of these 3 sets consists of 28 targets, including our highlighted candidates MYL3, CA3, MDH2, ETFA but also other interesting targets such as dystrophin (DMD) or actinin 2 (ACTN2). Presumably due to small sample sizes, not all of these protein profiles revealed statistically significant differences concordant for both sample types in group comparisons. Nevertheless, almost 85% of them show indeed in at least one cohort or sample type a protein profile trend supporting leakage of muscle-specific proteins into circulation.

Author’s improvements to the manuscript:

Regarding this point, we have now added a paragraph in the sub-section called “Protein profiles associated with muscular dystrophy” (under “Results”, page 10) and addressed this briefly in the fourth paragraph of the “Discussion” section (page 14).

Continuation Referee #1 (Remarks):

5. Finally, I would like the authors to comment on the likely abundance of the proteins that are biomarkers in the study and on assay specificity (the authors state that at least 50 events were counted on the Luminex assay - I am asking about the background in the assay). This comes from thinking about the assay itself, which was performed on a 500X dilution of the plasma or serum samples - that is, performed on 0.2% sample rather than 100% or 50% serum or plasma, which might have enabled less abundant proteins to be counted. The Uhlen lab has studied their antibodies exhaustively, and they have a clear idea about peptide cross-reactivity; is this in part the reason for the use of 0.2% sample?

Author’s response:

Regarding the latter part of this comment, namely the choice of sample dilution rate, we would like to recall that the single-binder assay setup we employed here allows a multiplex protein profiling in directly labelled and non-fractionated or depleted serum/plasma. This protocol has been previously optimized by us where samples were labelled at a 1:10 dilution rate and further diluted 1:50 in an assay buffer prior to incubation with an antibody bead array (Schwenk *et al*, 2008). Other colleagues have published an optimized labelling protocol for serum proteomics based on a very similar total

dilution rate (1:450), where samples are labelled at 1:45 dilution rate and further diluted 1:10 prior to application on antibody arrays (Ingvarsson *et al*, 2007; Wingren *et al*, 2007). As we and the other colleagues have shown, a final dilution rate in this range allows for a multiplex profiling of both high and medium abundant classical plasma/serum proteins but also relatively low abundant tissue leakage products.

Profiling the relatively low abundant proteins is closely related to the analytical sensitivity, as also meant in the initial part of the referee's comment. Although we are constantly working on improving the sensitivity of detection of our assay platform, our currently reported lower limit of detections are in the range of 1 ng/ml (Schwenk *et al*, 2010b). This allows well for detection of proteins expected to be in plasma such as CA3 or CK. Furthermore, it also allows for analysis of tissue leakage products as these are expected to be present in plasma within a concentration range of approximately 0.1-500 ng/ml (Anderson *et al*, 2002). Thus, circulating tissue leakage products are expected to be differentially detected on our platform despite a lack of existing information on normal range abundance of these proteins in literature.

Author's improvements to the manuscript:

We have addressed this point in the third paragraph of our "Discussion" section (page 13-14).

Continuation Referee #1 (Remarks):

Recommendation: I would accept this paper for publication and I would ask the authors to discuss the five points I listed above. The paper is important for an underserved community of patients and families.

Author's comment:

As pointed by the referee, the findings of this study would indeed be of great interest and importance for the affected community, their families, physicians and researchers involved in the muscular dystrophy field. Yet, it will for sure raise an interest beyond this audience since it illustrates an approach relevant for many other rare diseases, where sample availability is a limiting aspect and an impediment for the identification of blood-based protein markers.

Referee #2 (Comments on Novelty/Model System):

Manuscript EMM-2013-03724 is a well executed study that has established several serum-associated muscle proteins as potential biomarkers for relatively rare genetic disorders, the group of X-linked muscular dystrophies.

Referee #2 (Remarks):

Manuscript EMM-2013-03724 by Cristina Al-Khalili Szigyarto and co-workers entitled 'Profiling geographically dispersed cohorts reveals protein markers in blood for Duchenne Muscular dystrophies' describes a well-executed screening study of a large and diverse cohort of blood samples from patients afflicted with X-linked muscular dystrophies. The antibody-based approach is well designed and properly presented, and appears to be a highly suitable way to discover novel protein biomarker candidates in serum samples. Since there is an urgent need for reliable biomarkers to improve diagnosis and prognosis of dystrophic patients, as well as determine the value of experimental treatments, this study is an important contribution to the field of muscular dystrophy research.

Author's comment:

The presented study, as mentioned by the Referee 2 has indeed revealed a handful concordant protein profiles in serum and plasma of muscular dystrophy patients and non-diseased controls associated with disease severity and clinical parameters, demonstrating the utility of this approach for identification of protein markers in a rare and genetic diseases such as muscular dystrophies. As

highlighted by the referee, this outcome has been achieved as a result of a highly collaborative and committed execution of the study within the framework of the EU-FP7 BIO-NMD project.

Continuation Referee #2 (Remarks):

To clarify a few points in this manuscript, the following issues should be addressed by the authors:

(i) One of the key findings of the antibody-based proteomic screening exercise presented in this paper is the elevated concentration of the CA3 isoform of carbonic anhydrase, which is a relatively common and abundant protein associated with muscle tissue that also appears to exhibit a fibre type-specific distribution. The authors are encouraged to better outline these points in their discussion with respect to:

- Potential muscular dystrophy-unrelated effects: Is it possible that CA3 levels in blood samples are majorly affected by certain crush injuries, mild forms of exercise, a variety of drug effects or pathologies unrelated to muscular dystrophy? It is certainly well known that both exercise or injury trigger elevated serum CA3 levels. Please see: Brancaccio P, Lippi G, Maffulli N. Biochemical markers of muscular damage. *Clin Chem Lab Med.* 2010 Jun;48(6):757-67.

Author's response:

The point raised by the referee concerning potential non-dystrophy related changes of CA3 serum levels, in particular due to other pathologies, physical exercise or ageing, is highly relevant to address especially since biomarkers presently used for clinical management of DMD, like CK, exhibit this disadvantage. CA3 is involved in several cellular processes like fatty acid metabolism, mitochondrial ATP synthesis and oxidative stress response and has been linked to several disorders like epilepsy, osteoporosis, gastric or neurological disorders. Consequently, the activity of CA3 is targeted by several drugs, which modulate its activity rather than its expression and abundance in serum and plasma. A thorough investigation of factors that affect CA3 blood levels would indeed add value not only to the use of CA3 as a biomarker but also to the biological understanding of CA3 expression changes in humans.

Author's improvements to the manuscript:

The effect of various pathologies not related to muscular dystrophy such as drugs, ageing and physical exercise is now addressed in the "Discussion" section (page 15-16).

Continuation Referee #2 (Remarks):

- Fibre type specific distribution of CA3: A previous immunocytochemical and biochemical study suggests that cytosolic carbonic anhydrase isoform CA3 is present at higher levels in soleus muscle, as compared to deep vastus lateralis muscle and superficial vastus lateralis muscle, which are composed of predominantly type I, IIa, and IIb fibres, respectively. Please see: Frémont P, Charest PM, Côté C, Rogers PA. Carbonic anhydrase III in skeletal muscle fibres: an immunocytochemical and biochemical study. *J Histochem Cytochem.* 1988 Jul;36(7):775-82. In addition, chronic electrostimulation-induced transitions from fast to slower contracting muscle phenotypes is associated with marked increase in CA3 mRNA levels, agreeing with the concept of a fibre type-specific distribution of this enzyme in skeletal muscle. Please see: Brownson C, Isenberg H, Brown W, Salmons S, Edwards Y. Changes in skeletal muscle gene transcription induced by chronic stimulation. *Muscle Nerve.* 1988 Nov;11(11):1183-9. This issue of the fibre type distribution of the novel biomarker candidate CA3 should be outlined in more detail and it should be discussed whether this fact may have an effect on the pathobiochemical leakage of this protein in muscular dystrophy.

Author's response:

The comment raised by the referee regarding tissue and muscle fibre type distribution of CA3 requires careful evaluation. CA3 distribution in different tissues and cells might explain the origin of proteins leaking into the bloodstream and the mechanism by which this occurs. A CA3 accumulation in blood might reflect deterioration of skeletal muscles enriched in muscle fibre type I, such as the soleus muscle, due to its fibre type specific expression (Brancaccio et al. 2010 and Shima et al. 1983). Accumulation of CA3 in blood of DMD patients can be a consequence of type I muscle fibre replacement with connective tissue but also increased expression of CA3 due to continuous stimulation of the muscles.

Author's improvements to the manuscript:

The implications of CA3 distribution in different muscles and fibre types as a muscle wasting biomarker are addressed now in the "Discussion" section (page 15-16).

Continuation Referee #2 (Remarks):

- *Additional reference on CA3 suitability: Besides the already quoted references on CA3, a previous paper by Väänänen and co-workers has studied CA3 leakage as a potential biomarker of type I skeletal muscle fibres in polymyositis, muscular dystrophies, amyotrophic lateral sclerosis and other neurogenic diseases. Please see: Väänänen HK, Takala TE, Tolonen U, Vuori J, Myllylä VV. Muscle-specific carbonic anhydrase III is a more sensitive marker of muscle damage than creatine kinase in neuromuscular disorders. Arch Neurol. 1988 Nov;45(11):1254-6.*

Author's improvements to the manuscript:

In agreement with the referee's comment regarding the specificity of CA3 as a marker for muscular dystrophy, we have introduced appropriate references and extended the discussion of this point in the "Discussion" section (page 15-16).

Continuation Referee #2 (Remarks):

(ii) *Standardization of sampling procedure: Analytical studies involving blood, serum or plasma samples involve crucial steps of specimen handling, fractionation and storage. Do the authors have an indication how critical the choice of tubes, material, chemicals, centrifugation steps, handling time and storage method is for the production of highly reproducible findings?*

Author's response:

The referee is indeed correct to point out the importance of pre-analytical factors on sample integrity and their potential effect on downstream analysis. In a recent study (Qundos *et al*, 2013), we investigated some of these important parameters such as post-centrifugation delay and temperature with regard to alterations in protein profiles generated on antibody bead arrays. Although we identified few proteins sensitive to differences in the pre-analytical chain, we are well aware that certain classes of proteins such as cytokines are degraded at a much higher pace than proteins of cellular leakage. Therefore, one of the success factors for all but especially those studies built upon the analysis of multi-centre sample collections is to use standardized protocols for sample collection, handling and storage with full documentation. Thus, the serum and plasma samples analysed in this study were collected at four different clinical sites according to a collection protocol adopted within the BIO-NMD consortium.

Nevertheless, the presented study, in addition to two previous studies of ours (Schwenk *et al*, 2010a; Qundos *et al*, 2013) and a recent one by others (O Neal *et al*, 2014) reveals that certain proteins can be detected differentially even in serum and plasma samples collected from same individual at the same clinical site. As shown in **Supporting Information Figure 4** and the new **Supporting Information Figure 5.A**, we observed that the protein profiles across all plasma and serum samples collected at different sites grouped mainly by the blood preparation type. Furthermore, despite the standardized sample collection, handling and storage protocols utilized within the BIO-NMD

consortium, we observed a slight effect of sample origin. As can be seen in the new **Supporting Information Figure 5.B**, the plasma samples originating from Ferrara, Italy varied slightly from the ones collected in London or Newcastle, UK. Such a difference might be explained due to exposure of the specimen to temperature fluctuations during transit.

In conclusion, despite the utmost attention and awareness, it is difficult to retain identical conditions both during the collection and the transfer of samples. We observed indications of such effects and therefore addressed this by performing individual comparative analysis of data generated for each sample preparation type and originating from each clinical site for the identification of concordant differential protein profiles. In contrast to a majority of protein biomarker discovery studies, the findings we have presented in this work are therefore not constrained by a certain blood preparation type originating from a single sample collection site.

Author's improvements to the manuscript:

Regarding this point, we have now added the **Supporting Information Figure 5** and emphasized the importance of using standardized protocols for sample collection, handling and storage in the second paragraph of the "Discussion" section (page 13).

Continuation Referee #2 (Remarks):

It is known from large-scale proteomic studies of serum and plasma samples from human blood that even small changes in the protocol can have substantial effects on the measurement of protein markers. In this respect, would the implementation of the suggested new serum biomarkers require a common international protocol independent of national guidelines to establish a proper screening system for patients with dystrophinopathies.

Author's response:

In the presented study we analysed serum and EDTA plasma, the two most commonly used blood preparation types and the differential protein profiles we have highlighted as candidates were revealed in sample collections collected at more than one site and both in plasma and serum. Therefore, the findings we presented in this work revealed by analysis of geographically dispersed cohorts are not constrained by a certain blood preparation type and are more robust than outcome of any possible studies relying on analysis of a single sample type from a single location. However, subsequent analyses of larger sample cohorts could indeed unveil whether any of these two blood preparations is preferable. For a future implementation of a marker or marker panel, particular sample collection protocols and guidelines might therefore be needed to be assessed and defined to achieve best and most robust assay performance. Such guidelines could then of course be applied across different cohorts and international study sites.

Author's improvements to the manuscript:

Regarding this point, we have now added a discussion point towards the end of our "Discussion" section (page 18).

Continuation Referee #2 (Remarks):

(iii) The authors are encouraged to discuss in more detail the actual robustness of their newly suggested assay system, detailing the best usage of biomarker panels, the most suitable protein marker signature for differential analyses of dystrophinopathies, and the critical issue of sensitivity versus specificity.

Author's response:

This is indeed a very relevant and important point. As we emphasized in our discussion, the presented discoveries from our screening efforts require further verification as well as translation into assay systems that can be used in a clinical environment. This not only includes to develop a

clinically robust test in a sandwich immunoassay format, but also to challenge the clinical sensitivity of this test with new and independent sets of samples. While the latter is a challenge for rare diseases, we have yet recently shown a path of successfully translating “discovery” assays into clinically more applicable tests (Qundos *et al*, 2014). This would include i) collecting commercially available mono- and poly-clonal antibodies for targets of interest, ii) generating monoclonal antibodies for targets towards which no commercial antibodies are available iii) epitope-mapping of both the commercially available antibodies and the antibodies generated within the Human Protein Atlas on high-density peptide arrays (Buus *et al*, 2012) and iv) testing multi-antibody sandwich assays to identify matching pairs of these antibodies with distinct epitopes revealing a good assay sensitivity.

From a clinical point of view, the most urgent and important need is to monitor disease progression in dystrophinopathies. Besides, there are also intermediate cases of DMD/BMD patients where genetic tests do not provide conclusive answers facilitating the clinical management of the disease. Therefore, although all the eleven candidates we have highlighted deserve to be investigated in follow-up studies, initially a panel consisting of CA3, MDH2, MYL3, TNNT3 and ETFA could be prioritized. As summarized in **Table 2**, profiles for this set of 5 proteins would allow for assessment of both the DMD/BMD and the ambulation status. Several well-characterized antibodies targeting these 5 targets could be collected and/or generated and matching pairs of capture and detection antibodies for each target could be identified allowing for a 5-plex sandwich assay, which could be further challenged with new sample material.

Author’s improvements to the manuscript:

Regarding this point, we have now added a paragraph towards the end of our “Discussion” section (page 18).

Continuation Referee #2 (Remarks):

(iv) Although this study represents a large-scale proteomic style study that is based on high-throughput antibody technology, the paper does not mention or discuss in much detail the large volume of previous proteomic studies, that certainly have a certain degree of relevance to this screening exercise. For a recent comprehensive review of the proteomic analysis of muscle specimens, please see: Holland et al. Proteomics of the dystrophin-glycoprotein complex and dystrophinopathy. Curr Protein Pept Sci. 2013 Oct 4. [Epub ahead of print] PMID: 24106963. This review also contains a discussion on the very recent publication of the muscle secretome from mdx muscle cells by Partridge and co-workers, which might be of interest to the general discussion on leakage versus secretion of distinct protein populations in pathological muscles. Please see: Duguez S, Duddy W, Johnston H, Lainé J, Le Bihan MC, Brown KJ, Bigot A, Hathout Y, Butler-Browne G, Partridge T. Dystrophin deficiency leads to disturbance of LAMP1-vesicle-associated protein secretion. Cell Mol Life Sci. 2013 Jun;70(12):2159-74.

Author’s response:

In agreement with the referee’s comments we acknowledge these previous proteomics efforts aiming to characterize the effect of dystrophin absence on the muscle proteome. Several publications have scrutinized the proteome of *mdx* model organisms and DMD patients providing valuable results, which indirectly have enabled our study (Holland et al. 2013). The targets included in our study were selected based on literature mining and prioritization of experimental evidence used by the Pathway Studio software. To emphasize this further, the description of the target selection has been now revised also to address the first comment by Referee#1 .

The proteomics studies regarding the process by which cytoplasmic muscle proteins enter the blood stream as a consequence of absent dystrophin gives an additional perspective that requires careful attention. Duguez, Partridge and colleagues have shown that proteins are secreted in the culturing media by dystrophin-deficient myotubes during cultivation. The secreted myotubes are likely to be release into the blood stream and serve as biomarker candidates. Among the markers identified in our work, MYL3 and MDH2 has been shown to be secreted via the lysosomal vesicle trafficking whereas CA3 not (Duguez et al 2012). Thus the origin and release of each marker into the blood has to be scrutinized by further analysis.

Author's improvements to the manuscript:

The relevant studies mentioned by the referee are now cited in the fourth paragraph of the "Discussion" section (page 14). Regarding release of identified markers from the muscle tissues, additional text has been included in the fifth paragraph of "Discussion" (page 15).

Continuation Referee #2 (Remarks):

(v) *Potentially relevant references on the proteomic analysis of mdx plasma: Two previous studies have focused on the evaluation of potential changes in the plasma from the dystrophic mdx animal model of Duchenne muscular dystrophy. The authors should evaluate whether any findings of these studies correlate to their analysis of human blood samples and whether fundamental differences exist between the mild phenotype of the mdx mouse and the highly progressive form of human DMD. Please see: Alagaratnam, S.; Mertens, B.J.; Dalebout, J.C.; Deelder, A.M.; van Ommen, G.J.; den Dunnen, J.T.; 't Hoen, P.A. Serum protein profiling in mice: identification of Factor XIIIa as a potential biomarker for muscular dystrophy. Proteomics, 2008, 8, 1552-1563; and Colussi, C.; Banfi, C.; Brioschi, M.; Tremoli, E.; Straino, S.; Spallotta, F.; Mai, A.; Rotili, D.; Capogrossi, M.C.; Gaetano, C. Proteomic profile of differentially expressed plasma proteins from dystrophic mice and following suberoylanilide hydroxamic acid treatment. Proteomics Clin. Appl., 2010, 4, 71-83.*

Author's response:

The studies mentioned by the referee are indeed relevant. In fact, these previous reports on proteomics findings in *mdx* organisms have been considered during selection of the targets. Yet, due to lack of suitable antibodies not all published targets were included in our study (see revised description of target and antibody selection under Materials & Methods in the sub-section of "Selection of candidate targets and design of the antibody array" on page 21-22 and under Results in the sub-section of "Study and experimental design" on page 5-6).

Author's improvements to the manuscript:

We have now addressed this point by adding a section to the "Discussion" section and by citing the relevant references (page 16 and 17)

Continuation Referee #2 (Remarks):

(vi) *Human muscle biopsies: If sufficient numbers of patient muscle samples are available, potential changes in the novel markers could be evaluated on the level of muscle tissue in order to possibly determine whether distinct differences exist between disease-associated sarcolemmal leakage and natural cellular secretion processes. Confocal microscopy would only require very small samples, if available, to determine potential changes in the subcellular location and/or overall concentration of the new markers within fibres.*

Author's response:

We agree with the referee that a detailed analysis of muscle tissue and the sub-cellular location of the candidate proteins is of high interest and value. Although we believe such an effort is beyond the scope of the presented work, we are very much interested in initiating and being involved in such analyses as follow-up studies which though require strictly controlled collection of matched tissue and blood samples for such dedicated investigation. While collection of muscle tissue material from affected young children is a major challenge, we are participating in on-going international initiatives (such as IRDiRC), which are putting an effort in biobanking and enabling access of muscle tissue material.

Author's improvements to the manuscript:

In the final part of our discussion we have now emphasized the importance of analysing by which mechanism each and one of the identified markers is released into the blood stream (page 19).

Continuation Referee #2 (Remarks):

(vii) Minor point: In the References list, 5 references are missing volume or page numbers. This includes the References: Fagerberg et al..., Mercuri and Muntoni..., Moat et al..., Nishita et al..., and Staunton et al...!

Author's improvements to the manuscript:

We have now added the missing volume and/or page numbers for these references.

Referee #3 (Remarks):

This paper is contains a lot of data on the profiling of proteins expressed in 345 blood samples from DMD patients collected from 4 different clinical centres. The data show significant differences in four proteins between patients and controls which correlate with the age in sub cohorts of patients and with the clinical severity as evidenced from the data on BMD patients.

This is a disease where biomarkers are badly needed to monitor the progress of patients in clinical trials and these data would need to be validated by other groups. As far as I know, Somalogics using a different technology are the only other investigators to study this as extensively.

Author's comment:

It would be a great opportunity to disseminate our findings, as also stated by the referee, so that we and other researchers in the field can initiate even more extensive studies by joining efforts in collecting new sample material and performing follow-up studies to investigate the identified candidates in a more dedicated way.

Continuation Referee #3 (Remarks):

The data are of high quality and interesting. They are more valuable than most data sets because the patients are collected from four clinical centres.

Author's comment:

We appreciate this positive feedback about the value of the information we generated within this study. As highlighted by the referee, our approach demonstrates the great potential of identifying and verifying candidate plasma/serum proteins by using several independent sample collections. The presented findings are indeed of special interest for the muscular dystrophy field but the key approach of joining efforts starting at the level of a synchronized sample collection at various clinics represents an important example for the possibility of broad-scale protein profiling for identification of serum/plasma markers in rare diseases.

Continuation Referee #3 (Remarks):

It is difficult to predict whether these data will be confirmed by others. The authors relate their observations to plausible mechanisms but I might have expected to observe more than one protein derived from the same pathway of the disease process. Certainly network analysis would add more significance to the data.

These results are important. However, since the four protein markers are independent markers I think further work is needed to support their validity as biomarkers in DMD.

Author's response:

We absolutely agree with the referee that further steps are needed to assure the value of the discovered markers (please also refer to our previous replies above) and this is one of the motivations why the current markers are preferably called candidates.

Regarding the view about the independence of the identified candidates, we would like to first recall that the target selection phase of this study involved extensive pathway analysis in addition to other complementary approaches as we elaborated now in the revised version of our manuscript in the sub-section of "Selection of candidate targets and design of the antibody array" (under Materials & Methods, page 21-22) and in the sub-section of "Study and experimental design" (under Results, page 5-6). Thus, the target list utilized in this study intrinsically comprises a large portion of proteins known to be interacting with each other. To exemplify this, we have now included **Supporting Information Figure 12** showing the outcome of STRING analysis where the eleven targets listed in **Table 2** were used as input to obtain an interaction network with up to 25 interactors for these 11 targets with a confidence score of 0.8. Although not all of them directly and closely interacting with each other, these eleven targets (circled in red) and other targets that were included in the study but revealed statistically less significant differential profiles (circled in green), are not entirely unrelated. From a disease mechanistic perspective, one may sure argue for why distinct candidates have been found and not a collection of proteins interacting within a certain pathway or known to be involved in a certain cellular process. However, reasons for not covering all these targets originates from (i) the analytical accessibility of these post leakage or secretion, (ii) the performance of the used antibodies to target these, as well as (iii) the sensitivity of the method. Therefore, we believe that all the eleven highlighted candidates originating or not originating from a single pathway is not an indicator for the validity of these candidates. We also would like to refer to the 4th point raised by Referee#1, which is closely related to this point. As we clarified there, although a large portion of the muscle-specific proteins revealed a "DMD increased" and/or "BMD increased" trend as compared to the control groups, only a small portion of these protein profiles revealed statistically significant differences concordant across different sample collections and blood preparation types. Based on other in-house analyses and communications with biostatisticians and epidemiologists, we would like to emphasize the challenge of lack of statistical power in the analysis due to small sample sizes in rare diseases and the effect of combining samples collected at different sites. In conclusion, we believe that the indications we provide here will allow narrowing down the number of candidate proteins for coming studies that for sure should include protein targets within dedicated pathways or networks involving them.

REFERENCES

- Anderson NL & Anderson NG (2002) The human plasma proteome: history, character, and diagnostic prospects. *Mol Cell Proteomics* **1**: 845–867
- Brancaccio P, Lippi G & Maffulli N (2010) Biochemical markers of muscular damage. *Clin Chem Lab Med* **48**: 757–767
- Buus S, Rockberg J, Forsström B, Nilsson P, Uhlén M & Schafer-Nielsen C (2012) High-resolution Mapping of Linear Antibody Epitopes Using Ultrahigh-density Peptide Microarrays. *Molecular & Cellular Proteomics* **11**: 1790–1800
- Colwill K, Renewable Protein Binder Working Group & Gräslund S (2011) A roadmap to generate renewable protein binders to the human proteome. *Nat Methods* **8**: 551–558
- Duguez S, Duddy W, Johnston H, Lainé J, Le Bihan MC, Brown KJ, Bigot A, Hathout Y, Butler-Browne G & Partridge T (2013) Dystrophin deficiency leads to disturbance of LAMP1-vesicle-associated protein secretion. *Cell Mol Life Sci* **70**: 2159–2174 DOI 10.1007/s00018-012-1248-2
- Holland A, Carberry S & Ohlendieck K (2013) Proteomics of the dystrophin-glycoprotein complex and dystrophinopathy. *Curr Protein Pept Sci* **14**: 680–697
- Ingvarsson J, Larsson A, Sjöholm AG, Truedsson L, Jansson B, Borrebaeck CAK & Wingren C (2007) Design of recombinant antibody microarrays for serum protein profiling: targeting of complement proteins. *J Proteome Res* **6**: 3527–3536
- O'Neal WK, Anderson W, Basta PV, Carretta EE, Doerschuk CM, Barr RG, Bleecker ER, Christenson SA, Curtis JL, Han MK, Hansel NN, Kanner RE, Kleerup EC, Martinez FJ, Miller BE, Peters SP, Rennard SI, Scholand MB, Tal-Singer R, Woodruff PG, et al (2014) Comparison of

serum, EDTA plasma and P100 plasma for luminex-based biomarker multiplex assays in patients with chronic obstructive pulmonary disease in the SPIROMICS study. *J Transl Med* **12**: 9

Qundos U, Hong M-G, Tybring G, Divers M, Odeberg J, Uhlén M, Nilsson P & Schwenk JM (2013) Profiling post-centrifugation delay of serum and plasma with antibody bead arrays. *J Proteomics* **95**: 46–54

Qundos U, Johannesson H, Fredolini C, O’Hurley G, Branca R, Uhlén M, Wiklund F, Bjartell A, Nilsson P, Schwenk JM (2014) Analysis of plasma from prostate cancer patients links decreased carnosine dipeptidase 1 levels to lymph node metastasis. *Translation Proteomics* **2**: 14-24

Schwenk JM, Gry M, Rimini R, Uhlén M & Nilsson P (2008) Antibody suspension bead arrays within serum proteomics. *J Proteome Res* **7**: 3168–3179

Schwenk JM, Igel U, Kato BS, Nicholson G, Karpe F, Uhlén M & Nilsson P (2010a) Comparative protein profiling of serum and plasma using an antibody suspension bead array approach. *Proteomics* **10**: 532–540

Schwenk JM, Igel U, Neiman M, Langen H, Becker C, Bjartell A, Ponten F, Wiklund F, Grönberg H, Nilsson P & Uhlén M (2010b) Toward next generation plasma profiling via heat-induced epitope retrieval and array-based assays. *Molecular & Cellular Proteomics* **9**: 2497–2507

Shima K, Tashiro K, Hibi N, Tsukada Y & Hirai H (1983) Carbonic anhydrase-III immunohistochemical localization in human skeletal muscle - Springer. *Acta Neuropathol* **59**: 237-239

Stoevesandt O & Taussig MJ (2012) Affinity proteomics: the role of specific binding reagents in human proteome analysis. *Expert Rev Proteomics* **9**: 401–414

Wingren C, Ingvarsson J, Dexlin L, Szul D & Borrebaeck CAK (2007) Design of recombinant antibody microarrays for complex proteome analysis: choice of sample labeling-tag and solid support. *Proteomics* **7**: 3055–3065

3rd Editorial Decision

29 April 2014

Thank you for the submission of your revised manuscript to EMBO Molecular Medicine. We have now received the enclosed reports from the referees who were asked to re-assess it. As you will see the reviewers are now supportive and I am pleased to inform you that we will be able to accept your manuscript pending (editorial) final amendments.

Please submit your revised manuscript within two weeks. I look forward to seeing a revised form of your manuscript.

***** Reviewer's comments *****

Referee #1 (Remarks):

I read the revised manuscript carefully, and I am happy to suggest that the paper is accepted as written. The authors did everything I asked of them, and did it well.

Referee #2 (Remarks):

The authors have addressed all my points raised about the original submission. The revised paper is now an excellent contribution to the field of biomarker discovery for rare diseases.

Referee #3 (Remarks):

The authors have thoroughly revised the paper and responded well to all the points raised. I recommend publication.